# Distribution-Informed Neural Networks for Domain Adaptation Regression

**Jun Wu, Jingrui He, Sheng Wang, Kaiyu Guan, Elizabeth Ainsworth**
University of Illinois Urbana-Champaign
{junwu3,jingrui,sheng12,kaiyug,ainswort}@illinois.edu

## Abstract

In this paper, we study the problem of domain adaptation regression, which learns a regressor for a target domain by leveraging the knowledge from a relevant source domain. We start by proposing a distribution-informed neural network, which aims to build distribution-aware relationship of inputs and outputs from different domains. This allows us to develop a simple domain adaptation regression framework, which subsumes popular domain adaptation approaches based on domain invariant representation learning, reweighting, and adaptive Gaussian process. The resulting findings not only explain the connections of existing domain adaptation approaches, but also motivate the efficient training of domain adaptation approaches with overparameterized neural networks. We also analyze the convergence and generalization error bound of our framework based on the distribution-informed neural network. Specifically, our generalization bound focuses explicitly on the maximum mean discrepancy in the RKHS induced by the neural tangent kernel of distribution-informed neural network. This is in sharp contrast to the existing work which relies on domain discrepancy in the latent feature space heuristically formed by one or several hidden neural layers. The efficacy of our framework is also empirically verified on a variety of domain adaptation regression benchmarks.

## 1 Introduction

Domain adaptation tackles the knowledge transfer from a source domain with adequate label information to a relevant target domain with little or no label information. It is shown [5, 39, 11, 13] that the generalization performance on the target domain can be improved by leveraging the source domain knowledge in both classification [60] and regression [12] settings.

In this paper, we focus on studying the domain adaptation regression problem, as classification can be naturally formulated as regression [34]. In the past decades, most domain adaptation approaches were developed using the following paradigms: learning domain invariant representation [42, 43], reweighting the source examples [11, 13, 30], or deriving adaptive transfer kernel for Gaussian process [7, 44]. More recently, modern practice for domain adaptation repeatedly demonstrates the benefit of considering overparameterized neural networks as the backbones for feature extraction [1, 37, 20, 47, 51]. In this case, a $L$-layer neural network would be manually divided into two parts: the feature extraction function (e.g., the first $l$ layers) and the prediction function (e.g., the left $L - l$ layers). Then, the core idea of domain adaptation with overparameterized neural networks is to enhance the adaptation of source and target domains in the hidden feature space learned by the feature extraction layers. Nevertheless, it is unclear how those overparameterized neural networks affect the convergence and generalization of domain adaptation approaches during model training. Moreover, the heuristic selection of feature extraction layers might lead to sub-optimal performance of domain adaptation approaches in practice, as the lower and higher layers of an overparameterized neural network might encode different information [58].

36th Conference on Neural Information Processing Systems (NeurIPS 2022).

Compared with standard supervised learning, the key challenge of domain adaptation lies in the distribution shift between source and target domains. This indicates that the relationship between inputs and outputs might be different across domains. As illustrated in Figure 1, due to the distribution shift across domains, two similar input data points (e.g., $x_2 \approx x_3$) might have significantly different outputs (i.e., $y_2 \neq y_3$). This is also observed in recent work [52]. To solve this problem, we propose a distribution-informed neural network to build the unified relationship of inputs and outputs from different domains. This neural network can be explained as the integration of standard feature representation learning and input-oriented distribution representation learning. Then, a simple domain adaptation regression framework named DINO is proposed based on the distribution-informed neural network, followed by the theoretical analysis on its convergence and generalization performance.

In particular, we show that the distribution discrepancy of source and target domains can be defined by the evolution of distribution-informed neural network during training. That is, we measure the distribution shift using the maximum mean discrepancy (MMD) [24] in the reproducing kernel Hilbert space (RKHS) induced by the neural tangent kernel of distribution-informed neural network. Compared to previous works [37, 20, 1], our domain discrepancy measure has the following benefits. First, it can be estimated from the entire neural network, whereas previous works usually estimate the domain discrepancy in the pre-defined feature extraction layers. For example, [37] extracts features of input source and target examples and then estimates the discrepancy using MMD with standard kernels (e.g., RBF kernel). Intuitively, it is a two-fold composition kernel of the neural kernel induced by the feature extractor and the standard kernel. Second, our domain discrepancy measure focuses on the training dynamics of the neural network (e.g., $f_{\theta+\Delta\theta}(x) - f_\theta(x)$). This is in sharp contrast to the previous works which measure the distribution shift using the static state of the neural network (e.g., $f_\theta(x)$). As shown in Figure 1,

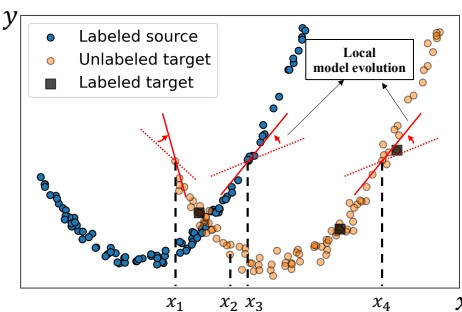

Figure 1: Illustration of the domain adaptation regression, where $x_1, x_3$ denotes two source data points and $x_2, x_4$ denotes two target data points. Here solid and dotted lines denote two consecutive states of the neural network around data points during training. (1) Similar data points (e.g., $x_2$ and $x_3$ from different domains) might have different outputs. (2) Source data point $x_3$ is more likely to be aligned with target data point $x_4$, as they have similar local model evolution.

a source data point $x_3$ might have similar output with both $x_1$ and $x_4$ from target domain, when simply considering the trained neural network $f_\theta(x)$. But we see that the neural network has similar evolution patterns at $x_3$ and $x_4$ but different patterns at $x_3$ and $x_1$. That is, $x_3$ is more likely be aligned with $x_4$ after distribution minimization across domains, That explained why the training dynamics of the neural network can better identify the distribution shift across domains.

In addition, we show that in special cases, the training dynamic of the distribution-informed neural network can also explain the rationale of the existing domain adaptation techniques. To be more specific, we have the following observations. (1) At random initialization, our distribution-informed neural network is equivalent to the adaptive Gaussian process [7, 44]. (2) Under gradient descent training, our domain adaptation framework based on the distribution-informed neural network can recover the prediction function of the reweighting domain adaptation approaches [30, 11, 13]. (3) When the distribution representation is shared by all the examples, our framework degenerates into standard domain-invariant representation learning [37, 20, 1]. Compared to previous works, our contributions can be summarized as follows.

- *Unified Framework:* We propose a simple domain adaptation regression framework DINO based on the distribution-informed neural network, and then identify its connections to previous techniques, including domain invariant representation learning, reweighting and adaptive Gaussian process.

- *Theoretical Results:* It is shown that for the distribution-informed neural network with sufficient width, the convergence and generalization of our framework can be theoretically guaranteed. Different from previous deep domain adaptation theories, we propose to measure the distribution shift across domains using the training dynamics of the neural network.

- *Empirical Performance:* We experimentally investigate the performance of our DINO framework on a variety of domain adaptation regression benchmarks, and show its effectiveness over state-of-the-art baselines.

The rest of this paper is organized as follows. We introduce the related work in Section 2, and summarize the preliminaries of previous domain adaptation techniques in Section 3. Section 4 shows the proposed domain adaptation regression framework, followed by its theoretical analysis. The experimental results are presented in Section 5. Finally, we conclude the paper in Section 6.

## 2 Related Work

### 2.1 Domain Adaptation

Domain adaptation [5, 52, 53, 62, 63] improves the generalization performance of a learning algorithm on the target domain, by leveraging the knowledge from a relevant source domain with adequate labeled data. There are three solutions to bridge the distribution gap between source and target domains. (1) *Domain invariant representation:* It maps the source and target examples into a new feature space such that the distribution discrepancy between source and target domains can be explicitly minimized [42, 37, 43, 20, 61, 60, 10]. (2) *Reweighting:* The core idea is to correct the difference between distributions by multiplying the prediction loss of each source example by a non-negative weight [30, 14, 11, 13, 47]. (3) *Gaussian process:* It generalized the standard Gaussian process regression to the domain adaptation setting by learning an adaptive transfer kernel [7, 44, 38, 50]. By using the neural networks as the backbones, modern domain adaptation techniques have achieved state-of-the-art performance in a variety of real-world tasks [28, 8]. Nevertheless, little effort has been devoted to theoretically analyzing the domain adaptation techniques with overparameterized neural networks.

More recently, it has been revealed [10] that there is a gap between domain adaptation regression and domain adaptation classification problems. That is, one common loss function of domain adaptation regression approaches is mean square error (MSE), whereas domain adaptation classification approaches [37, 20] often uses cross-entropy loss with softmax. Specifically, softmax changes the feature scales (i.e., Frobenius norm of feature matrix $||H||_F$ where $H$ is the hidden feature learned by the neural network), and the change of feature scales might lead to the performance degradation of domain adaptation [10]. Therefore, in this paper, we focus on the neural network with MSE loss for domain adaptation regression. Different from the existing domain adaptation regression approaches, e.g., domain invariant representation learning [10], reweighting [16], and adaptive Gaussian process [7], we develop the domain adaptation regression algorithms based on a novel distribution-informed neural network and derive the theoretical analysis regarding the convergence and generalization bound for our algorithms.

### 2.2 Overparameterized Neural Networks

It has been revealed from the perspective bias-variance decomposition [56] that the generalization error of neural networks decreases with respect to the model complexity (e.g., number of neurons). The benefit of increasing the number of neurons has also been empirically confirmed in modern neural networks [32, 59]. Moreover, the convergence and generalization of overparameterized neural networks have been studied in the neural tangent kernel (NTK) regime [31, 3, 4, 35, 17, 18, 2, 48, 9, 33]. However, those works focus on the standard supervised learning with the assumption that training and testing examples follow the same distribution. In real scenarios, this assumption often fails due to the distribution shift [38]. More recently, it is found that previous theoretical results can be generalized to both federated learning [29] and multi-task learning [45]. Our work fundamentally differs from them in that little label information is available in the target domain for our studied domain adaptation problem. This motivates us to explicitly analyze the domain discrepancy when using neural networks for domain adaptation.

## 3 Preliminaries

In this section, we introduce the notations and background of different domain adaptation techniques.

## 3.1 Notation and Neural Network Architecture

Let $\mathcal{X}$ and $\mathcal{Y}$ denote the input space and output space. Following [6], we assume that the joint probability distribution $\mathbb{P}$ of any domain is drawn from a probability distribution space $\mathscr{P}$ over $\mathcal{X} \times \mathcal{Y}$. For domain adaptation regression, we have a source domain with labeled examples $\{x_i^{\text{src}}, y_i^{\text{src}}\}_{i=1}^{n_{\text{src}}}$ drawn from joint probability distribution $\mathbb{P}^{\text{src}}$, and a target domain with both label examples $\{x_j^{\text{tgt}}, y_j^{\text{tgt}}\}_{j=1}^{n_{\text{tgt}}^l}$ from $\mathbb{P}^{\text{tgt}}$ and unlabeled examples $\{x_j^{\text{tgt}}\}_{j=1}^{n_{\text{tgt}}^u}$ $(n_{\text{tgt}}^l \ll n_{\text{tgt}}^u)$ drawn from marginal probability distribution $\mathbb{P}_X^{\text{tgt}}$. The goal is to predict the outputs of unlabeled target examples by leveraging the label information from the source domain. In this paper, we consider an $L$-layer fully-connected neural network $f_\theta(\cdot)$, which maps the input $x \in \mathcal{X} \subset \mathbb{R}^d$ into the output $y \in \mathcal{Y} \subset \mathbb{R}$. Here $\theta$ denotes all the parameters of $f(\cdot)$ (e.g., $\theta_0$ denotes the initialized parameters at time stamp $t = 0$) and $d$ denotes the dimensionality of input data. In next section, we generalize the standard fully-connected neural network $f_\theta(\cdot)$ to the distribution-informed neural network $\tilde{f}(\cdot)$ for domain adaptation. For notation simplicity, we let $X^{\text{src}} = \{x_i^{\text{src}}\}_{i=1}^{n_{\text{src}}}$ and $Y^{\text{src}} = \{y_i^{\text{src}}\}_{i=1}^{n_{\text{src}}}$ for source examples. We use similar notations $X_l^{\text{tgt}}$ and $Y_l^{\text{tgt}}$ for target examples. Then we use $X = X^{\text{src}} \cup X_l^{\text{tgt}}$ denote all the labeled training inputs, and $Y = Y^{\text{src}} \cup Y_l^{\text{tgt}}$ be the corresponding outputs.

## 3.2 Domain Adaptation Techniques

We first review the existing domain adaptation techniques. In Subsection 4.4, we show that these techniques can be explained in a unified framework.

### 3.2.1 Domain Invariant Representation

The key idea of domain invariant representation learning is to map the source and target examples into a new feature space where the distribution discrepancy across domains is explicitly minimized [20, 60]. The objective function of this framework can be summarized as follows.

$$\min_\theta \mathbb{E}_{(x,y)\sim\mathbb{P}^{\text{src}}} \left[ L(f_\theta(x), y) \right] + d\left( \mathbb{P}^{\text{src}}, \mathbb{P}^{\text{tgt}} \right) \tag{1}$$

where $d\left( \mathbb{P}^{\text{src}}, \mathbb{P}^{\text{tgt}} \right)$ denotes the domain discrepancy measure in the hidden feature space learned by $f_\theta(\cdot)$, and $L(\cdot, \cdot)$ is the loss function.

### 3.2.2 Reweighting

Reweighting aims to correct the difference between source and target distributions by reweighting the source examples, i.e., multiplying the prediction loss of every source example by a non-negative weight [14, 11, 13].

$$\min_\theta \mathbb{E}_{(x,y)\sim\mathbb{P}^{\text{src}}} \left[ w^{\text{src}}(x, y) \cdot L(f_\theta(x), y) \right] \tag{2}$$

where $w^{\text{src}}(x, y) = \frac{\mathbb{P}^{\text{tgt}}(x,y)}{\mathbb{P}^{\text{src}}(x,y)} \geq 0$ denotes the importance of every source example. The importance ratio $w^{\text{src}}(x, y)$ can be estimated using kernel mean matching (KMM) [30, 57] or adversarial learning [47, 16].

### 3.2.3 Adaptive Gaussian Process

Following [7], the domain adaptation regression problem can also be formulated as the Gaussian process with an adaptive transfer kernel. Specifically, the adaptive transfer kernel can be defined as

$$K' = \begin{bmatrix} K(X^{\text{src}}, X^{\text{src}}) & \tau \cdot K(X^{\text{src}}, X_l^{\text{tgt}}) \\ \tau \cdot K(X_l^{\text{tgt}}, X^{\text{src}}) & K(X_l^{\text{tgt}}, X_l^{\text{tgt}}) \end{bmatrix} \tag{3}$$

where $\tau \in [0, 1]$ is a trainable parameter explicitly indicating the relatedness of source and target domains, and $K(\cdot, \cdot)$ is a base kernel, e.g., RBF kernel. Then following the standard Gaussian process regression [49], the predictions of this adaptive Gaussian process over testing target data can be analytically derived.

# 4 A Unified Framework

In this section, we propose the distribution-informed neural network, and then present a simple domain adaptation regression framework.

## 4.1 Distribution-Informed Neural Network

An $L$-layer fully-connected neural network $f(\cdot)$ can be written as $f_\theta(x) = \phi_{\theta^{<L}}(x)^T w$ where $\theta^{<L}$ is the vector of parameters in the first $L-1$ layers and $w$ is the parameter of the output layer (we assume the output layer has no bias). For a single task, the neural network $f(\cdot)$ can model the relationship between input data and output label. But it might suffer in the domain adaptation scenarios due to the distribution shift. Thus, given a domain distribution $\mathbb{P} \in \mathscr{P}$ and an input example $x \sim \mathbb{P}$, we propose to explicitly learn the distribution-related output as $g_{w_g}(\mathbb{P}|x) = \Phi_x(\mathbb{P})^T w_g$ where $\Phi_x(\mathbb{P})$ is the input-oriented distribution feature representation, and $w_g$ is the parameter. Given finite basis examples $\tilde{x}_1, \cdots, \tilde{x}_n \sim \mathbb{P}$, $\Phi_x(\mathbb{P})$ can be formally defined as follows.

$$\Phi_x(\mathbb{P}) = \sum_{i=1}^n \beta_{x,\tilde{x}_i} \langle \cdot, \tilde{x}_i \rangle_{\mathcal{K}_\mathcal{X}} \tag{4}$$

where $\beta_{x,\tilde{x}_i} \in \mathbb{R}$ is the coefficient of $\Phi_x(\mathbb{P})$ in the space spanned by $\langle \cdot, \tilde{x}_i \rangle_{\mathcal{K}_\mathcal{X}}$ and indicates the similarity of input example $x$ and basis example $\tilde{x}_i$. Here $\langle \cdot, \tilde{x}_i \rangle_{\mathcal{K}_\mathcal{X}}$ denotes the feature map of $\tilde{x}_i$ in the NNGP kernel space induced by infinitely-wide $f(\cdot)$. Therefore, $\Phi_x(\mathbb{P})$ represents the distribution representation of $\mathbb{P}$ when an example $x$ is observed in the domain associated with sampling distribution $\mathbb{P}$.

One special case is when $\beta_{x,\tilde{x}_i} = 1/n$, it degenerates into the mean mapping of $\mathbb{P}$ in the kernel space, which is commonly used for measuring the distribution discrepancy/similarity [24]. Compared with this plain distribution representation $\frac{1}{n} \sum_{i=1}^n \langle \cdot, \tilde{x}_i \rangle_{\mathcal{K}_\mathcal{X}}$, $\Phi_x(\mathbb{P})$ pays more attention to the region around $x$ with large $\beta_{x,\tilde{x}_i}$ for $\tilde{x}_i \in \mathcal{X}$. For example, for data points $x_1$ and $x_4$ in Figure 1, they would share the same plain distribution representation, as they are sampled from the same target domain. But it is feasible to differentiate them using the parametric distribution representation $\Phi_x(\mathbb{P})$.

Generally, by taking both example $x$ and its associated probability $\mathbb{P}$ as random variables, Bayes' theorem tells us that $P(x, \mathbb{P}) = P(x) \cdot P(\mathbb{P}|x)$ where $P(\cdot)$ denotes the probability of an observed event. The event involving $x$ can be described by observing its representation $f(x)$; given $x$, the event involving $\mathbb{P}$ is the input-oriented distribution representation $\Phi_x(\mathbb{P})$. This motivates us to develop a distribution-informed neural network as follows.

$$\tilde{f}(x, \mathbb{P}) := f_\theta(x) \cdot g_{w_g}(\mathbb{P}|x) = \left(\phi_{\theta^{<L}}(x)^T w\right) \cdot \left(\Phi_x(\mathbb{P})^T w_g\right) = w^T \left(\phi_{\theta^{<L}}(x)\Phi_x(\mathbb{P})^T\right) w_g \tag{5}$$

The intuition behind Eq. (5) can be explained as follows. For domain adaptation, the feature representation of an input example $x$ is domain-dependent, as two similar examples ($x^{\text{src}} \approx x^{\text{tgt}}$) from different domains might have distinctive outputs ($y^{\text{src}} \neq y^{\text{tgt}}$). The input-oriented distribution representation $\Phi_x(\mathbb{P})$ allows us to identify the source and target examples with similar inputs but distinctive outputs. By analyzing the distribution-informed neural network at initialization and under gradient descent training, we propose two domain adaptation algorithms: DINO-INIT (see Subsection 4.2) and DINO-TRAIN (see Subsection 4.3), respectively.

## 4.2 Initialization

We start by studying the distribution-informed neural network at initialization, e.g., all the parameters of distribution-informed neural network $\tilde{f}(\cdot)$ are initialized as standard normal variables. Inspired by the connections between standard neural networks and Gaussian processes [15, 34, 22, 54], we show that the distribution-informed neural network $\tilde{f}(x, \mathbb{P})$ in Eq (5) is equivalent to a domain adaptive Gaussian process, which explains the existing domain adaption regression algorithm [7].

**Lemma 4.1.** *Assume that all the parameters of distribution-informed neural network $\tilde{f}(\cdot)$ are initialized as standard normal variables, when the network width goes to infinity, the output function of $\tilde{f}(x)$ in Eq (5) at initialization is iid centered Gaussian process, i.e., $\tilde{f}(\cdot) \sim \mathcal{N}\left(0, \mathcal{K}^{DA}\right)$ with*

$$\mathcal{K}^{DA}\left((x, \mathbb{P}), (x', \mathbb{P}')\right) = \mathcal{K}_\mathcal{X}(x, x') \cdot \mathcal{K}_{\mathscr{P}|\mathcal{X}}(\mathbb{P}, \mathbb{P}'|x, x')$$

*where $\mathcal{K}_\mathcal{X}(\cdot, \cdot)$ is the NNGP kernel induced by the neural network $f(\cdot)$ over input space $\mathcal{X}$, and $\mathcal{K}_\mathscr{P}(\cdot, \cdot)$ is a distribution kernel identifying the similarity of two input-oriented distributions $\mathbb{P}$ and $\mathbb{P}'$ over distribution space $\mathscr{P}$:*

$$\mathcal{K}_{\mathscr{P}|\mathcal{X}}\left(\mathbb{P}, \mathbb{P}'|x, x'\right) = \sum_{i=1}^n \sum_{j=1}^{n'} \beta_{x,\tilde{x}_i} \beta_{x',\tilde{x}'_j} \mathcal{K}_\mathcal{X}(\tilde{x}_i, \tilde{x}'_j) \tag{6}$$

*where $n$ ($n'$) is the number of basis examples sampled from the distribution $\mathbb{P}$ ($\mathbb{P}'$).*

This lemma motivates us to propose an adaptive Gaussian process regression algorithm named `DINO-INIT` with adaptive transfer kernel defined as $\mathcal{K}^{DA}$. For notation simplicity, we represent the distribution-informed kernel $\mathcal{K}^{DA}((x,\mathbb{P}),(x',\mathbb{P}'))$ as $\mathcal{K}^{DA}(x,x')$ in the following. In this case, we consider a noisy model $y_i^r = \tilde{f}(x_i^r, \mathbb{P}^r) + \epsilon_i^r$ ($r \in \{\text{src}, \text{tgt}\}$), where the noise $\epsilon_i^r$ follows a zero-mean Gaussian $\mathcal{N}(0, \sigma_r^2)$. Given labeled training examples $(X, Y)$, we assume a prior Gaussian process $p(Y) = \mathcal{N}(0, \mathcal{K}^{DA})$ where $\mathcal{K}^{DA}(X, X)$ is a block matrix. That is, $\mathcal{K}^{DA}(X, X) = \begin{bmatrix} \mathcal{K}_{11}^{DA} & \mathcal{K}_{12}^{DA} \\ \mathcal{K}_{21}^{DA} & \mathcal{K}_{22}^{DA} \end{bmatrix}$ where $\mathcal{K}_{11}^{DA} := \mathcal{K}^{DA}(X^{\text{src}}, X^{\text{src}})$ ($\mathcal{K}_{22}^{DA} := \mathcal{K}^{DA}(X_l^{\text{tgt}}, X_l^{\text{tgt}})$) denotes the kernel matrix of source (target) data, and $\mathcal{K}_{12}^{DA} = (\mathcal{K}_{21}^{DA})^T := \mathcal{K}^{DA}(X^{\text{src}}, X_l^{\text{tgt}})$ denotes the kernel matrix across domains. Then for testing target examples $X_*^{\text{tgt}}$, their output can be inferred using the predictive distribution $p(Y | X_*^{\text{tgt}}, X^{\text{src}}, X_l^{\text{tgt}}) = \mathcal{N}(\bar{\mu}, \bar{\Sigma})$. Similar to standard Gaussian process regression [49], the mean and variance of the predictive distribution can be exactly calculated as follows.

$$\bar{\mu} = \mathcal{K}^{DA}(X_*^{\text{tgt}}, X)C^{-1}Y \qquad \bar{\Sigma} = \mathcal{K}^{DA}(X_*^{\text{tgt}}, X_*^{\text{tgt}}) - \mathcal{K}^{DA}(X_*^{\text{tgt}}, X)C^{-1}\mathcal{K}^{DA}(X_*^{\text{tgt}}, X)^T$$

where $C = \mathcal{K}^{DA}(X, X) + \begin{bmatrix} \sigma_{\text{src}}^2 \mathbb{I}_{n_{\text{src}}} & 0 \\ 0 & \sigma_{\text{tgt}}^2 \mathbb{I}_{n_{\text{tgt}}^l} \end{bmatrix}$. Here $\mathbb{I}$ denotes the identity matrix.

This adaptive Gaussian process involves the following parameters: coefficient $\beta_{x,x_i}$ in the kernel function $\mathcal{K}^{DA}$ and noise variance $\sigma_{\text{src}}, \sigma_{\text{tgt}}$. Following [7], we optimize these parameters by maximizing the conditional likelihood $p(Y_l^{\text{tgt}} | X_l^{\text{tgt}}, X^{\text{src}}, Y^{\text{src}})$. It can be seen that $p(Y_l^{\text{tgt}} | X_l^{\text{tgt}}, X^{\text{src}}, Y^{\text{src}})$ is also a Gaussian process, i.e., $p(Y_l^{\text{tgt}} | X_l^{\text{tgt}}, X^{\text{src}}, Y^{\text{src}}) = \mathcal{N}\left( \mathcal{K}_{21}^{DA} \left( \mathcal{K}_{11}^{DA} + \sigma_{\text{src}}^2 \mathbb{I}_{n_{\text{src}}} \right)^{-1} Y^{\text{src}}, \left( \mathcal{K}_{22}^{DA} + \sigma_{\text{tgt}}^2 \mathbb{I}_{n_{\text{tgt}}^l} \right) - \mathcal{K}_{21}^{DA} \left( \mathcal{K}_{11}^{DA} + \sigma_{\text{src}}^2 \mathbb{I}_{n_{\text{src}}} \right)^{-1} \mathcal{K}_{12}^{DA} \right)$.

**Instantition of the coefficient** $\beta_{x,\tilde{x}_i}$: It is notable that it can only optimize the coefficient $\beta_{x,\tilde{x}_i}$ of Eq. (4) for training (source or target) examples. The coefficient of testing target examples $X_*^{\text{tgt}}$ is unknown. To solve this problem, in this paper, we formulate the estimate of $\beta_{x^r,\tilde{x}_i^r}$ ($r \in \{\text{src}, \text{tgt}\}$) as $\beta_{x,\tilde{x}_i^r} = [x^r \circ \tilde{x}_i^r]^T w^r$, where $w^r$ is a domain-specific parameter vector and $\circ$ denotes vector concatenation. Note that domain adaptation assumes that the domain labels of all input examples are known. As shown in Eq. (4), the input-oriented distribution representation can be learned from the domain-specific vector $w^r$ and the basis examples $\tilde{x}_1^r, \cdots, \tilde{x}_{n_r}^r$ from domain $r$. Here $r$ is determined by the domain label of the input example $x$. As illustrated in Algorithm 1, we use all the training source (target) examples as the basis source (target) examples $\tilde{x}_1^r, \cdots, \tilde{x}_{n_r}^r$. As a result, the input-oriented distribution representation learning of testing target examples $X_*^{\text{tgt}}$ can be inferred by $w^{\text{tgt}}$ and the pre-defined basis target examples.

### 4.3 Gradient Descent Training

Here we assume that the model parameters of the distribution-informed neural network can be updated by gradient descent during training. Based on the distribution-informed neural network, we propose a novel `DINO-TRAIN` algorithm. Its objective function using mean square error is formulated as follows.

$$\mathcal{L}(\theta) = \frac{\alpha}{2n_{\text{src}}} \sum_{i=1}^{n_{\text{src}}} \left( \tilde{f}(x_i^{\text{src}}, \mathbb{P}^{\text{src}}) - y_i^{\text{src}} \right)^2 + \frac{1-\alpha}{2n_{\text{tgt}}^l} \sum_{j=1}^{n_{\text{tgt}}^l} \left( \tilde{f}(x_j^{\text{tgt}}, \mathbb{P}^{\text{tgt}}) - y_j^{\text{tgt}} \right)^2 + \frac{\mu}{2} \hat{\text{MMD}}_{\Theta_{DA}}^2 \left( \mathbb{P}^{\text{src}}, \mathbb{P}^{\text{tgt}} \right) \tag{7}$$

where $\alpha \in (0,1)$ and $\mu \geq 0$ are hyperparameters to balance different terms. The first two terms are standard supervised learning losses over labeled source and target examples, and the third one is the empirical maximum mean discrepancy (MMD) [24] in the RKHS $\mathcal{H}_{DA}$ induced by the neural tangent kernel of our distribution-informed neural network, i.e.,

$$\hat{\text{MMD}}_{\Theta_{DA}}^2 \left( \mathbb{P}^{\text{src}}, \mathbb{P}^{\text{tgt}} \right) = \left\| \frac{1}{n_{\text{src}}} \sum_{i=1}^{n_{\text{src}}} \nabla_\theta \tilde{f}(x_i^{\text{src}}, \mathbb{P}^{\text{src}}) - \frac{1}{n_{\text{tgt}}} \sum_{j=1}^{n_{\text{tgt}}} \nabla_\theta \tilde{f}(x_j^{\text{tgt}}, \mathbb{P}^{\text{tgt}}) \right\|_{\mathcal{H}_{DA}}^2 \tag{8}$$

where $n_{\text{tgt}} = n_{\text{tgt}}^l + n_{\text{tgt}}^u$ is the total number of target training examples. As shown in Theorem 4.5 below, our framework empirically minimizes the upper error bound of the expected prediction error in the target domain.

**Remark.** In our framework of Eq. (7), we measure the distribution shift using the training dynamics of the distribution-informed neural network over the source and target examples. This is fundamentally different from previous works [20, 60, 1] associated with a two-stage domain discrepancy measurement. That is, those works would first learn the feature representation in hidden neural layers and then measure the domain discrepancy in the learned feature space. In the second stage, they usually require additional modules, e.g., auxiliary neural networks [10, 21, 19] or pre-defined distribution distances [37]. Therefore, compared to previous works, our discrepancy measure Eq. (8) can not only unify the domain adaptation regression framework with neural networks, but also enable the theoretical convergence and generalization analysis in the following.

Before analyzing the dynamics of the distribution-informed neural network under Eq. (7), we first introduce the following assumption.

**Assumption 4.2.** Given labeled data $X = X^{\text{src}} \cup X_l^{\text{tgt}}$, we assume $\lambda_{\min}(\Theta(X,X)) > 0$ and $\lambda_{\min}(\mathcal{K}_{\mathscr{P}|\mathcal{X}}(X,X)) > 0$, where $\Theta(X,X) = \nabla_\theta f(X)\nabla_\theta f(X)^T$ is standard neural tangent kernel [31] induced by $f(\cdot)$ with infinite width, and $\mathcal{K}_{\mathscr{P}|\mathcal{X}}(X,X)$ is the input-oriented distribution kernel[1] (see Eq. (6)). Here, $\lambda_{\min}(\cdot)$ denotes the smallest eigenvalue.

The assumption $\lambda_{\min}(\Theta(X,X)) > 0$ has been studied in previous works [31, 18, 3]. It holds as long as no two inputs $x_i$ and $x_j$ are parallel in real scenarios. The implication of this assumption is that the input data (e.g., images) would not be linearly changed by a single factor, i.e., there is no two inputs such that $x_i = \varsigma \cdot x_j$ for some $\varsigma \in \mathbb{R}$.

**Lemma 4.3.** *Given the labeled training data $X = X^{src} \cup X_l^{tgt}$ and the basis examples $\{\tilde{x}_i^r\}_{i=1}^{\tilde{n}_r}$ ($r \in \{src, tgt\}$), we let $\Upsilon \in \mathbb{R}^{(n_{src}+n_{tgt}^l) \times (\tilde{n}_{src}+\tilde{n}_{tgt})}$ denote the coefficient matrix of $\mathcal{K}_{\mathscr{P}|\mathcal{X}}(X,X)$, where for $x_k \in X$ ($k = 1, \cdots, n_{src} + n_{tgt}^l$), its $k^{th}$ row $[\Upsilon]_{k,:} = [\beta_{x_k,\tilde{x}_1^{src}}, \cdots, \beta_{x_k,\tilde{x}_{\tilde{n}_{src}}^{src}}, 0, \cdots, 0]$ when $x_k \in X^{src}$, $[\Upsilon]_{k,:} = [0, \cdots, 0, \beta_{x_k,\tilde{x}_1^{tgt}}, \cdots, \beta_{x_k,\tilde{x}_{\tilde{n}_{tgt}}^{tgt}}]$ otherwise. Then, the assumption $\lambda_{min}(\mathcal{K}_{\mathscr{P}|\mathcal{X}}(X,X)) > 0$ holds when $\Upsilon^T\Upsilon$ is positive definite.*

Lemma 4.3 shows that in real scenarios, the assumption $\lambda_{\min}(\mathcal{K}_{\mathscr{P}|\mathcal{X}}(X,X)) > 0$ can be guaranteed by the positive definiteness of the coefficient matrix $\Upsilon$. We also empirically evaluate the smallest eigenvalue $\lambda_{\min}(\mathcal{K}_{\mathscr{P}|\mathcal{X}}(X,X))$ in Subsection 5.2.

**Theorem 4.4.** *For any coefficient $\beta_{x,\tilde{x}_i}$ of the input-oriented distribution representation in Eq. (4), there exists $\eta^* \in \mathbb{R}_+$ such that for the infinitely-wide distribution-informed neural network $\tilde{f}(\cdot)$ trained under gradient flow with learning rate $\eta < \eta^*$, the test prediction $\tilde{f}_{\theta_t}(X_*^{tgt})$ of the domain adaptation regression in Eq. (7) over test target data $X_*^{tgt}$ is*

$$\tilde{f}_{\theta_t}(X_*^{tgt}) = \tilde{f}_{\theta_0}(X_*^{tgt}) - \Theta_{DA}(X_*^{tgt}, X)\Theta_{DA}(X,X)^{-1}\left(\mathbb{I} - e^{-\eta\Theta_{DA}\tilde{C}t}\right)\left(\tilde{f}_{\theta_0}(X) - Y\right)$$

*where $\Theta_{DA}(\cdot,\cdot)$ is the distribution-informed NTK, i.e., $\Theta_{DA}(x,x') = \Theta(x,x') \cdot \mathcal{K}_{\mathscr{P}|\mathcal{X}}(\mathbb{P}, \mathbb{P}'|x,x')$ and $\tilde{C} = diag\{\underbrace{\alpha/n_{src}, \cdots, \alpha/n_{src}}_{n_{src}}, \underbrace{(1-\alpha)/n_{tgt}^l, \cdots, (1-\alpha)/n_{tgt}^l}_{n_{tgt}^l}\}$ is a diagonal matrix. Moreover, under the assumption 4.2, when the network width goes to infinity, $\lim_{t\to\infty}\tilde{f}_{\theta_t}(X_*^{tgt})$ converges to a Gaussian process with mean $\mu(X_*^{tgt})$ and variance $\Sigma(X_*^{tgt}, X_*^{tgt})$ as follows.*

$$\mu(X_*^{tgt}) = \Theta_{DA}(X_*^{tgt}, X)\Theta_{DA}(X,X)^{-1}Y$$
$$\Sigma(X_*^{tgt}, X_*^{tgt}) = \mathcal{K}^{DA}(X_*^{tgt}, X_*^{tgt}) + \Theta_{DA}(X_*^{tgt}, X)\Theta_{DA}(X,X)^{-1}\mathcal{K}^{DA}\Theta_{DA}(X,X)^{-1}\Theta_{DA}(X, X_*^{tgt})$$
$$- \left(\Theta_{DA}(X_*^{tgt}, X)\Theta_{DA}(X,X)^{-1}\mathcal{K}^{DA}(X, X_*^{tgt}) + h.c.\right)$$

*where "$+h.c.$" means "plus the Hermitian conjugate".*

In addition, our result on generalization bound is given in the following theorem.

**Theorem 4.5.** *Assume for any training example $(x,y) \in \mathcal{X} \times \mathcal{Y}$, we have $(\tilde{f}(x,\mathbb{P}) - y)^2 \leq M_0$ for some constant $M_0 \geq 0$. Let $\tilde{\mathcal{F}}$ be the hypothesis space induced by infinitely-wide neural networks $\tilde{f}$. Then, for any $\tilde{f} \in \tilde{\mathcal{F}}$ and $\delta > 0$, with probability at least $1 - \delta$, the expected error in the target*

---

[1] Here we let $\mathcal{K}_{\mathscr{P}|\mathcal{X}}(x,x') = \mathcal{K}_{\mathscr{P}|\mathcal{X}}(\mathbb{P}, \mathbb{P}'|x,x')$ for brevity.

*domain can be bounded as follows.*

$$\mathcal{E}_{\mathbb{P}^{tgt}}(\tilde{f}) \leq \frac{\alpha}{n_{src}} \sum_{i=1}^{n_{src}} \left( \tilde{f}(x_i^{src}, \mathbb{P}^{src}) - y_i^{src} \right)^2 + \frac{1-\alpha}{n_{tgt}^l} \sum_{j=1}^{n_{tgt}^l} \left( \tilde{f}(x_j^{tgt}, \mathbb{P}^{tgt}) - y_j^{tgt} \right)^2$$
$$+ 8\alpha M_0 \cdot MMD_{\Theta_{DA}}\left( \mathbb{P}^{src}, \mathbb{P}^{tgt} \right) + \Omega$$

*where* $\Omega = 2\alpha\Re_{n_{src}}(\mathcal{H}_{src}) + 2(1-\alpha)\Re_{n_{tgt}^l}(\mathcal{H}_{tgt}) + M\sqrt{\frac{(n_{src}+n_{tgt}^l)\log(1/\delta)}{2}}$, $\mathcal{H}_r = \{(x,y) \to (\tilde{f}(x^r, \mathbb{P}^r) - y^r)^2 : \tilde{f} \in \mathcal{F}\}$ *is a set of functions* ($r \in \{src, tgt\}$), $\Re_{n_r}(\mathcal{H}_r)$ *is the Rademacher complexity of* $\mathcal{H}_r$ *given* $n_r$ *examples, and* $M = \max\{\alpha M_0/n_{src}, (1-\alpha)M_0/n_{tgt}^l\}$.

Though domain adaptation theories [5, 39] have been generalized to deep learning scenarios in recent years, neural networks are simply considered as an expressive feature extractor in modern generalization errors [60]. Nevertheless, recent works [61, 36] reveal that the feature space learned by the neural networks might worsen the adaptation between source and target domains. As a comparison, Theorem 4.5 provides a unified generalization error for neural network based domain adaptation, as it directly measures the domain discrepancy over input source and target examples in the RKHS induced by the distribution-informed NTK $\Theta_{DA}$.

### 4.4 Discussion

#### 4.4.1 Gaussian Process at Initialization

One special case of distribution-informed neural network at initialization is that when the coefficient $\beta_{x,x_i} = 1/n\|\bar{\Phi}_x(\mathbb{P})\|_{\mathcal{K}_{\mathcal{X}}}$ and $\bar{\Phi}_x(\mathbb{P}) = \frac{1}{n}\sum_{i=1}^{n}\langle \cdot, x_i\rangle_{\mathcal{K}_{\mathcal{X}}}$, it can be seen that $\Phi_x(\mathbb{P}) = \frac{1}{n\bar{\Phi}_x(\mathbb{P})}\sum_{i=1}^{n}\langle \cdot, x_i\rangle_{\mathcal{K}_{\mathcal{X}}}$ is the normalized mean mapping [24] of data distribution $\mathbb{P}$ in the NNGP kernel space. Then, the distribution kernel $\mathcal{K}_{\mathscr{P}|\mathcal{X}}(\mathbb{P}, \mathbb{P}'|x, x')$ in Eq. (6) can be explained as the inner product of the mean mappings of $\mathbb{P}$ and $\mathbb{P}'$. Moreover, using the definition of maximum mean discrepancy (MMD) [24], it can be shown that $\mathcal{K}_{\mathscr{P}|\mathcal{X}}(\mathbb{P}, \mathbb{P}'|x, x') = \frac{c - \hat{\text{MMD}}^2_{\mathcal{K}_{\mathcal{X}}}(\mathbb{P},\mathbb{P}')}{2\Phi_x(\mathbb{P})\Phi_x(\mathbb{P}')}$, where $\hat{\text{MMD}}_{\mathcal{K}_{\mathcal{X}}}(\cdot)$ denotes the empirical MMD in the NNGP kernel space, and $c = \frac{1}{n^2}\sum_{i,j=1}^{n}\mathcal{K}_{\mathcal{X}}(x_i, x_j) + \frac{1}{n'^2}\sum_{i,j=1}^{n'}\mathcal{K}_{\mathcal{X}}(x_i', x_j')$. It implies that $\mathcal{K}_{\mathscr{P}|\mathcal{X}}(\mathbb{P}, \mathbb{P}'|x, x')$ is negatively correlated with the popular MMD estimator. MMD measures the distribution distance using the difference of mean mappings of distributions, whereas $\mathcal{K}_{\mathscr{P}|\mathcal{X}}(\mathbb{P}, \mathbb{P}'|x, x')$ measures the distribution similarity via the inner product of the mean mappings of the distributions $\mathbb{P}$ and $\mathbb{P}'$ in the NNGP kernel space.

The following corollary shows that the existing domain adaptive Gaussian process [7] can be explained as implicitly learning the distribution-informed representation in the kernel space.

**Corollary 4.6.** *With the assumptions in Lemma 4.1, when* $\beta_{x,\tilde{x}_i} = 1/\|\sum_{i=1}^{n}\langle \cdot, \tilde{x}_i\rangle_{\mathcal{K}_{\mathcal{X}}}\|$, *the Gaussian process induced by* $\tilde{f}(x)$ *at the initialization would be equivalent to the adaptive Gaussian process in [7] over NNKP kernel.*

#### 4.4.2 Fully Trained Model

When there exist some labled target examples, the objective function of the reweighting domain adaptation techniques (see Subsection 3.2.2) can be rewritten as follows.

$$\min_{\theta} \frac{\alpha}{2n_{\text{src}}} \sum_{i=1}^{n_{\text{src}}} w_i^{\text{src}} \left( f(x_i^{\text{src}}) - y_i^{\text{src}} \right)_2^2 + \frac{1-\alpha}{2n_{\text{tgt}}} \sum_{j=1}^{n_{\text{tgt}}^l} \left( f(x_j^{\text{tgt}}) - y_j^{\text{tgt}} \right)_2^2 \tag{9}$$

The following corollary holds that our framework Eq. (7) can recover the reweighting approach Eq. (9) in the function space.

**Corollary 4.7.** *With the assumption in Theorem 4.4, our framework Eq. (7) with distribution-informed neural network* $\tilde{f}(\cdot)$ *can recover the popular reweighting domain adaptation approach Eq. (9) in the function space.*

Besides, we show that when the distribution representation is shared by all the examples, our framework can be naturally degenerated into the domain-invariant representation learning [37].

| Methods | $C \to N$ | $C \to S$ | $N \to C$ | $N \to S$ | $S \to C$ | $S \to N$ | Avg. |
|---|---|---|---|---|---|---|---|
| NNGP [34] | $2.041_{\pm 0.001}$ | $1.823_{\pm 0.001}$ | $0.445_{\pm 0.002}$ | $0.624_{\pm 0.001}$ | $0.197_{\pm 0.002}$ | $0.459_{\pm 0.002}$ | $0.932$ |
| NTKGP [25] | $1.345_{\pm 0.002}$ | $1.227_{\pm 0.000}$ | $0.323_{\pm 0.002}$ | $0.529_{\pm 0.004}$ | $0.248_{\pm 0.001}$ | $0.425_{\pm 0.002}$ | $0.683$ |
| AT-GP [7] | $0.194_{\pm 0.005}$ | $0.259_{\pm 0.002}$ | $\mathbf{0.104}_{\pm \mathbf{0.001}}$ | $0.252_{\pm 0.005}$ | $0.118_{\pm 0.003}$ | $0.189_{\pm 0.006}$ | $0.186$ |
| TL-NTK [38] | $0.164_{\pm 0.001}$ | $\mathbf{0.231}_{\pm \mathbf{0.000}}$ | $0.124_{\pm 0.005}$ | $0.242_{\pm 0.002}$ | $0.125_{\pm 0.001}$ | $0.197_{\pm 0.004}$ | $0.181$ |
| DINO-INIT (ours) | $0.128_{\pm 0.001}$ | $0.233_{\pm 0.003}$ | $0.114_{\pm 0.002}$ | $\mathbf{0.227}_{\pm \mathbf{0.002}}$ | $\mathbf{0.112}_{\pm \mathbf{0.001}}$ | $\mathbf{0.181}_{\pm \mathbf{0.005}}$ | $\mathbf{0.166}$ |
| DINO-TRAIN (ours) | $\mathbf{0.127}_{\pm \mathbf{0.002}}$ | $0.240_{\pm 0.003}$ | $0.127_{\pm 0.000}$ | $0.243_{\pm 0.000}$ | $0.128_{\pm 0.001}$ | $0.194_{\pm 0.001}$ | $0.177$ |

Table 1: Results of domain adaptation regression on dSprites

| Methods | $RL \to RC$ | $RL \to T$ | $RC \to RL$ | $RC \to T$ | $T \to RL$ | $T \to RC$ | Avg. |
|---|---|---|---|---|---|---|---|
| NNGP [34] | $0.313_{\pm 0.001}$ | $0.438_{\pm 0.004}$ | $0.356_{\pm 0.005}$ | $0.515_{\pm 0.008}$ | $0.367_{\pm 0.001}$ | $0.324_{\pm 0.004}$ | $0.386$ |
| NTKGP [25] | $0.396_{\pm 0.001}$ | $0.365_{\pm 0.001}$ | $0.200_{\pm 0.007}$ | $0.390_{\pm 0.003}$ | $0.390_{\pm 0.000}$ | $0.354_{\pm 0.003}$ | $0.349$ |
| AT-GP [7] | $0.214_{\pm 0.011}$ | $0.209_{\pm 0.002}$ | $0.227_{\pm 0.010}$ | $0.198_{\pm 0.002}$ | $0.236_{\pm 0.000}$ | $0.249_{\pm 0.000}$ | $0.222$ |
| TL-NTK [38] | $0.206_{\pm 0.004}$ | $0.200_{\pm 0.002}$ | $0.213_{\pm 0.000}$ | $0.197_{\pm 0.000}$ | $0.226_{\pm 0.001}$ | $0.218_{\pm 0.000}$ | $0.210$ |
| DINO-INIT (ours) | $0.204_{\pm 0.001}$ | $\mathbf{0.185}_{\pm \mathbf{0.006}}$ | $\mathbf{0.207}_{\pm \mathbf{0.003}}$ | $\mathbf{0.182}_{\pm \mathbf{0.004}}$ | $\mathbf{0.218}_{\pm \mathbf{0.001}}$ | $\mathbf{0.212}_{\pm \mathbf{0.001}}$ | $\mathbf{0.201}$ |
| DINO-TRAIN (ours) | $\mathbf{0.193}_{\pm \mathbf{0.001}}$ | $0.194_{\pm 0.003}$ | $\mathbf{0.207}_{\pm \mathbf{0.003}}$ | $0.188_{\pm 0.002}$ | $0.226_{\pm 0.001}$ | $0.218_{\pm 0.001}$ | $0.204$ |

Table 2: Results of domain adaptation regression on MPI3D

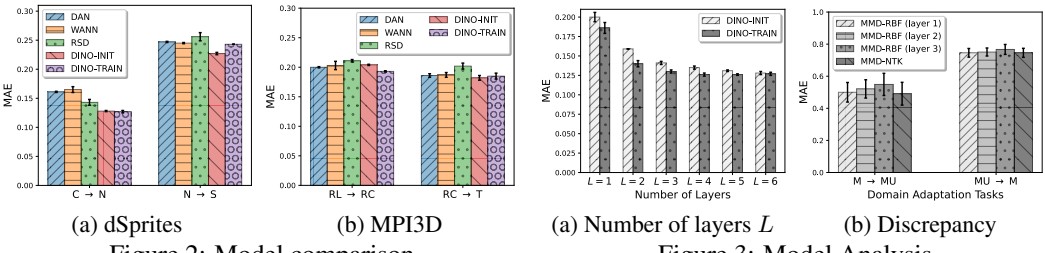

(a) dSprites  (b) MPI3D  (a) Number of layers $L$  (b) Discrepancy

Figure 2: Model comparison    Figure 3: Model Analysis

**Corollary 4.8.** *Under mild conditions, our framework Eq. (7) with distribution-informed neural network $\tilde{f}(\cdot)$ can recover the standard domain invariant representation learning in Eq. (1), where the domain discrepancy measure $d(\cdot, \cdot)$ is instantiated with MMD in RKHS induced by standard NTK $\Theta$.*

## 5    Experiments

**Data Sets:** We use three domain adaptation regression benchmarks. Following [10], we use two image data sets: dSprites [40] and MPI3D [23]. Specifically, dSprites has 737,280 images from three domains: Color (C), Noisy (N) and Scream (S). For each image, it has three regression tasks, i.e., predicting three factors of variations (scale, position X and Y). MPI3D contains over 3M images from three domains: Toy (T), Realistic (RC) and Real (RL). It is shown [10] that for each image, it involves two regression tasks, i.e., predicting three factors of variations (position X and Y). In addition, we also use a plant phenotyping data set. It predicts diverse traits (e.g., Nitrogen) of plants related to the plants' growth using leaf hyperspectral reflectance. Here we consider the following two domains [46]: Maize (M) and Maize_UNL (MU). In our case, the task is to predict the Nitrogen content of maize using the leaf hyperspectral reflectance (see Appendix A.11 for more details).

**Baselines:** In the experiments, we consider the following baseline methods. (1) Plain Gaussian process: NNGP [34] and NTKGP [25]. (2) Domain adaptive Gaussian processes: AT-GP [7] and TL-NTK [38]. (3) Deep domain adaptation methods: DAN [37], WANN [16] and RSD [10].

**Implementations:** In the experiments, our algorithms are implemented using a $L$-layer ($L = 6$) fully-connected neural network with ReLU (see Appendix A.11 for more details). The induced NNGP and neural tangent kernels induced can be estimated using the Neural Tangents package [41]. In addition, we set $\alpha = 0.5$ and $\mu = 0.1$ for DINO-TRAIN.

### 5.1    Results

Table 1, Table 2 and Table 3 provide the domain adaptation regression results on dSprites, MPI3D and Plant Phenotyping data sets. Following [10], we report the Mean Absolute Error (MAE) between the predicted outputs and the ground-truth outputs in the target test set (the best results are indicated in bold). It is observed that our method DINO achieves competitive performance over

the Gaussian process baselines. The weakly-trained `DINO-INIT` slightly outperforms `DINO-TRAIN` in some cases. This observation is consistent with previous work [33]. The relatively unstable performance of `DINO-TRAIN` might be caused by the large covariance of Gaussian process with NTK [25]. In addition, we also compare `DINO` with domain adaptation methods instantiated by overparameterized neural networks. WANN [16] reweights the source examples, and DAN [37] and RSD [10] learn the domain invariant representation. The performance comparison in Figure 2 confirms the effectiveness of our `DINO` approach over those state-of-the-art baselines.

| Methods | $M \rightarrow MU$ | $MU \rightarrow M$ |
|---|---|---|
| NNGP [34] | $0.562_{\pm 0.001}$ | $0.672_{\pm 0.010}$ |
| NTKGP [25] | $0.562_{\pm 0.004}$ | $0.702_{\pm 0.010}$ |
| AT-GP [7] | $\mathbf{0.308_{\pm 0.006}}$ | $0.593_{\pm 0.025}$ |
| TL-NTK [38] | $0.316_{\pm 0.008}$ | $0.488_{\pm 0.027}$ |
| `DINO-INIT` (ours) | $0.316_{\pm 0.007}$ | $0.645_{\pm 0.017}$ |
| `DINO-TRAIN` (ours) | $0.314_{\pm 0.009}$ | $\mathbf{0.443_{\pm 0.030}}$ |

Table 3: Results on Plant Phenotyping

### 5.2 Analysis

**Ablation Study:** We investigate the impact of input-oriented distribution representation in our `DINO` framework. We consider the following variants. `DINO-INIT` (`DINO-TRAIN`) w/o distribution feature: it would not utilize the input-oriented distribution representation, i.e., $g_{w_g}(\mathbb{P}|x) = 1$ for all $x \in \mathcal{X}$. `DINO-INIT` (`DINO-TRAIN`) w uniform distribution feature: the coefficient $\beta_{x,\tilde{x}_i} = 1/n$ is a constant. The results are

| Methods | $C \rightarrow N$ |
|---|---|
| `DINO-INIT` w/o distribution feature | $0.189_{\pm 0.002}$ |
| `DINO-INIT` w uniform distribution feature | $0.171_{\pm 0.001}$ |
| `DINO-INIT` | $0.128_{\pm 0.001}$ |
| `DINO-TRIAN` w/o distribution feature | $0.161_{\pm 0.002}$ |
| `DINO-TRIAN` w uniform distribution feature | $0.147_{\pm 0.001}$ |
| `DINO-TRIAN` | $0.127_{\pm 0.002}$ |

Table 4: Ablation study on dSprites

given in Table 4. We observe that the distribution representation improves the domain adaptation performance. Compared to globally shared distribution representation with $\beta_{x,\tilde{x}_i} = 1/n$, our parametric $\Phi_x(\mathbb{P})$ in Eq. (4) encourages to learn the local distribution representation around the input data point $x$. The empirical results in Table 4 confirmed the efficacy of the input-oriented distribution representation.

**Impact of Model Architecture and Discrepancy Measure:** Figure 3a shows the results of `DINO` with different number of neural layers $L$ on dSprites ($C \rightarrow N$). The results indicate that the performance of `DINO` is positively related to the expressiveness of NTK and NNGP kernel induced by the neural network. In addition, we compare the proposed $\hat{\text{MMD}}_{\Theta_{DA}}(\cdot, \cdot)$ with MMD-RBF (i.e., MMD with standard RBF kernel) in domain adaptation regression, where MMD-RBF is estimated in the feature space learned by different neural layers (see Appendix A.11 for more details). As shown in Figure 3b, the results on Plant Phenotyping data set reveal that the selection of feature space affects the model performance when using MMD-RBF, and our proposed $\hat{\text{MMD}}_{\Theta_{DA}}(\cdot, \cdot)$ outperforms the MMD-RBF.

**Positive Definiteness of $\mathcal{K}_{\mathscr{P}|\mathcal{X}}(X, X)$:** We report the smallest eigenvalue of the distribution kernel $\mathcal{K}_{\mathscr{P}|\mathcal{X}}(X, X)$ over the training examples on Plant Phenotyping data set. The smallest eigenvalues of $\mathcal{K}_{\mathscr{P}|\mathcal{X}}(X, X)$ in `DINO-TRAIN` after training are $3e-6$ and $8e-7$ on $M \rightarrow MU$ and $MU \rightarrow M$, respectively. This empirically confirms the positive definiteness of $\mathcal{K}_{\mathscr{P}|\mathcal{X}}(X, X)$ in `DINO-TRAIN`.

## 6 Conclusion

In this paper, we study the domain adaptation regression by proposing a distribution-informed neural network. Our domain adaptation framework with distribution-informed neural network subsumes the existing adaptation approaches based on domain invariant representation, reweighting, and adaptive Gaussian process. We also analyze the convergence and generalization bound of our framework. The experiments demonstrate the effectiveness of our `DINO` framework over state-of-the-art baselines.

## Acknowledgments and Disclosure of Funding

This work is supported by National Science Foundation under Award No. IIS-1947203, IIS-2117902, IIS-2137468, and Agriculture and Food Research Initiative (AFRI) grant no. 2020-67021-32799/project accession no.1024178 from the USDA National Institute of Food and Agriculture. The views and conclusions are those of the authors and should not be interpreted as representing the official policies of the funding agencies or the government.

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
