# A Appendix

In the appendix, we have the following results.

- In Appendix A.1, we summarize the main notations used in this paper.
- In Appendix A.2 - A.9, we show all the proofs of our theoretical results.
- In Appendix A.10, we present the overall training procedures (e.g., pseudo code) of our proposed DINO-INIT and DINO-TRAIN algorithms, as well as the limitations of our work.
- In Appendix A.11, we present additional data description and implementation details.

## A.1 Notation

| Notation | Definition |
|---|---|
| $\mathcal{X}, \mathcal{Y}, \mathscr{P}$ | Input, output and probability distribution space |
| $\mathbb{P}^{\text{src}}, \mathbb{P}^{\text{tgt}} \in \mathscr{P}$ | Source and target joint distributions |
| $\mathbb{P}_X^{\text{src}}, \mathbb{P}_X^{\text{tgt}}$ | Source and target marginal distributions |
| $\{x_i^{\text{src}}, y_i^{\text{src}}\}_{i=1}^{n_{\text{src}}}$ | Labeled source training examples |
| $\{x_j^{\text{tgt}}, y_j^{\text{tgt}}\}_{j=1}^{n_{\text{tgt}}^l}$ | Labeled target training examples |
| $\{x_j^{\text{tgt}}\}_{j=1}^{n_{\text{tgt}}^u}$ | Unlabeled target training examples |
| $X_*^{\text{tgt}}$ | Target testing examples |
| $X^{\text{src}} = \{x_i^{\text{src}}\}_{i=1}^{n_{\text{src}}}, Y^{\text{src}} = \{y_i^{\text{src}}\}_{i=1}^{n_{\text{src}}}$ | Labeled source input and output sets |
| $X_l^{\text{tgt}} = \{x_j^{\text{tgt}}\}_{j=1}^{n_{\text{tgt}}^l}, Y_l^{\text{tgt}} = \{y_j^{\text{tgt}}\}_{j=1}^{n_{\text{tgt}}^l}$ | Labeled target input and output sets |
| $X = X^{\text{src}} \cup X_l^{\text{tgt}}, Y = Y^{\text{src}} \cup Y_l^{\text{tgt}}$ | All the labeled training inputs, and the corresponding outputs |
| $\tilde{x}_i$ | Basis example of input-oriented distribution representation learning |
| $f(\cdot)$ | $L$-layer fully-connected neural network |
| $\tilde{f}(\cdot)$ | Distribution-informed neural network |
| $\mathcal{K}_{\mathcal{X}}(\cdot, \cdot)$ | NNGP kernel over $\mathcal{X}$ |
| $\mathcal{K}_{\mathscr{P}|\mathcal{X}}(\cdot, \cdot)$ | Distribution kernel over $\mathscr{P}$ |
| $\mathcal{K}^{DA}(\cdot, \cdot)$ | Distribution-informed NNGP kernel |
| $\Theta(\cdot, \cdot)$ | Neural tangent kernel (NTK) |
| $\Theta_{DA}(\cdot, \cdot)$ | Distribution-informed NTK |

Table 5: Notation

## A.2 Proof of Lemma 4.1

**Lemma 4.1.** Assume that all the parameters of $\tilde{f}(\cdot)$ follows standard normal distribution, in the limits as the layer width $d \to \infty$, the output function of the distribution-informed neural network $\tilde{f}(x)$ in Eq (5) at initialization is iid centered Gaussian process, i.e., $\tilde{f}(\cdot) \sim \mathcal{N}\left(0, \mathcal{K}^{DA}\right)$ where

$$\mathcal{K}^{DA}\left((x, \mathbb{P}), (x', \mathbb{P}')\right) = \mathcal{K}_{\mathcal{X}}(x, x') \cdot \mathcal{K}_{\mathscr{P}|\mathcal{X}}(\mathbb{P}, \mathbb{P}'|x, x')$$

where $\mathcal{K}_{\mathcal{X}}(\cdot, \cdot)$ is the NNGP kernel induced by the neural network $f(\cdot)$ over input space $\mathcal{X}$, and $\mathcal{K}_{\mathscr{P}}(\cdot, \cdot)$ is a distribution kernel identifying the similarity of the distributions $\mathbb{P}$ and $\mathbb{P}'$ over distribution space $\mathscr{P}$:

$$\mathcal{K}_{\mathscr{P}|\mathcal{X}}\left(\mathbb{P}, \mathbb{P}'|x, x'\right) = \sum_{i=1}^{n} \sum_{j=1}^{n'} \beta_{x, \tilde{x}_i} \beta_{x', \tilde{x}_j} \mathcal{K}_{\mathcal{X}}(\tilde{x}_i, \tilde{x}_j') \tag{10}$$

where $n$ ($n'$) is the number of examples in the domain associated with distribution $\mathbb{P}$ ($\mathbb{P}'$).

*Proof.* Following [34], it can be shown that the output function of a fully-connected neural networks $f(\cdot)$ is an iid centered Gaussian process with zero mean and variance $\mathcal{K}_{\mathcal{X}}(x, x')$ (i.e., NNGP kernel). Using definition of distribution-informed neural network $\tilde{f}(x)$ in Eq (5), we have $\mathbb{E}[\tilde{f}(x, \mathbb{P})] = 0$ (due to $\mathbb{E}[w] = 0$), and $\mathcal{K}^{DA}((x, \mathbb{P}), (x', \mathbb{P}')) = \mathbb{E}[\phi(x)^T \phi(x')] \cdot \Phi_x(\mathbb{P})^T \Phi_{x'}(\mathbb{P}')$ (due to $\mathbb{E}[w^T w] = 1$). Using the definition of $\Phi_x(\mathbb{P})$ in Eq. (4), we have $\mathcal{K}_{\mathscr{P}|\mathcal{X}}(\mathbb{P}, \mathbb{P}'|x, x') = \sum_{i=1}^{n} \sum_{j=1}^{n'} \beta_{x, \tilde{x}_i} \beta_{x', \tilde{x}_j'} \mathcal{K}_{\mathcal{X}}(\tilde{x}_i, \tilde{x}_j')$. $\square$

## A.3 Proof of Lemma 4.3

**Lemma 4.3.** Given the labeled training data $X = X^{\text{src}} \cup X_l^{\text{tgt}}$ and the basis examples $\{\tilde{x}_i^r\}_{i=1}^{\tilde{n}_r}$ ($r \in \{\text{src}, \text{tgt}\}$), we let $\Upsilon \in \mathbb{R}^{(n_{\text{src}}+n_{\text{tgt}}^l) \times (\tilde{n}_{\text{src}}+\tilde{n}_{\text{tgt}})}$ denote the coefficient matrix of $\mathcal{K}_{\mathscr{P}|\mathcal{X}}(X, X)$, where for $x_k \in X$ ($k = 1, \cdots, n_{\text{src}} + n_{\text{tgt}}^l$), its $k^{\text{th}}$ row $[\Upsilon]_{k,:} = [\beta_{x_k, \tilde{x}_1^{\text{src}}}, \cdots, \beta_{x_k, \tilde{x}_{\tilde{n}_{\text{src}}}^{\text{src}}}, 0, \cdots, 0]$ when $x_k \in X^{\text{src}}$, $[\Upsilon]_{k,:} = [0, \cdots, 0, \beta_{x_k, \tilde{x}_1^{\text{tgt}}}, \cdots, \beta_{x_k, \tilde{x}_{\tilde{n}_{\text{tgt}}}^{\text{tgt}}}]$ otherwise. Then, the assumption $\lambda_{\min}(\mathcal{K}_{\mathscr{P}|\mathcal{X}}(X, X)) > 0$ holds when $\Upsilon^T \Upsilon$ is positive definite.

*Proof.* Using the definition of the distribution kernel in Eq. (6), we have $\mathcal{K}_{\mathscr{P}|\mathcal{X}}(X, X) = \Upsilon \mathcal{K}_{\mathcal{X}}(\tilde{X}, \tilde{X}) \Upsilon^T$ where $\tilde{X}$ denotes all the basis examples. Then, we have $\lambda_{\min}(\mathcal{K}_{\mathscr{P}|\mathcal{X}}(X, X)) = \lambda_{\min}(\Upsilon \mathcal{K}_{\mathcal{X}}(\tilde{X}, \tilde{X}) \Upsilon^T) \geq \lambda_{\min}(\mathcal{K}_{\mathcal{X}}(\tilde{X}, \tilde{X})) \cdot \lambda_{\min}(\Upsilon^T \Upsilon)$. Here $\mathcal{K}_{\mathcal{X}}(\tilde{X}, \tilde{X})$ is the NNGP kernel matrix of the basis examples induced by the neural network $f(\cdot)$ with infinite width. It is shown [4] that the key difference between NNGP kernel and NTK is that NTK is generated by a fully-trained neural network, whereas NNGP kernel is produced by a weakly-trained neural network. That is, NNGP kernel is a special case of NTK when training only the output layer. Following [18, 3], when there is no two parallel inputs in $\tilde{X}$, we have $\lambda_{\min}(\mathcal{K}_{\mathcal{X}}(\tilde{X}, \tilde{X})) > 0$. Therefore, when $\Upsilon^T \Upsilon$ is positive definite, the assumption $\lambda_{\min}(\mathcal{K}_{\mathscr{P}|\mathcal{X}}(X, X)) > 0$ can hold. $\qquad \square$

## A.4 Proof of Theorem 4.4

**Theorem 4.4.** For any coefficient $\beta_{x, x_i}$ of the input-oriented distribution representation in Eq. (4), there exists $\eta^* \in \mathbb{R}_+$ such that for the infinitely-wide distribution-informed neural network $\tilde{f}(\cdot)$ trained under gradient flow with learning rate $\eta < \eta^*$, the test prediction $\tilde{f}_{\theta_t}(X_*^{\text{tgt}})$ of the domain adaptation regression in Eq. (7) over test target data $X_*^{\text{tgt}}$ is

$$\tilde{f}_{\theta_t}(X_*^{\text{tgt}}) = \tilde{f}_{\theta_0}(X_*^{\text{tgt}}) - \Theta_{DA}(X_*^{\text{tgt}}, X) \Theta_{DA}(X, X)^{-1} \left( \mathbb{I} - e^{-\eta \Theta_{DA} \tilde{C} t} \right) \left( \tilde{f}_{\theta_0}(X) - Y \right)$$

where $\Theta_{DA}(\cdot, \cdot)$ is the distribution-informed NTK, i.e., $\Theta_{DA}(x, x') = \Theta(x, x') \cdot \mathcal{K}_{\mathscr{P}|\mathcal{X}}(\mathbb{P}, \mathbb{P}'|x, x')$ and $\tilde{C} = \text{diag}\{\underbrace{\alpha/n_{\text{src}}, \cdots, \alpha/n_{\text{src}}}_{n_{\text{src}}}, \underbrace{(1-\alpha)/n_{\text{tgt}}^l, \cdots, (1-\alpha)/n_{\text{tgt}}^l}_{n_{\text{tgt}}^l}\}$ is a diagonal matrix. Moreover, under the assumption 4.2, when the network width goes to infinity, $\lim_{t \to \infty} \tilde{f}_{\theta_t}(X_*^{\text{tgt}})$ converges to a Gaussian process with mean $\mu(X_*^{\text{tgt}})$ and variance $\Sigma(X_*^{\text{tgt}}, X_*^{\text{tgt}})$ as follows.

$$\mu(X_*^{\text{tgt}}) = \Theta_{DA}\left(X_*^{\text{tgt}}, X\right) \Theta_{DA}(X, X)^{-1} Y$$
$$\Sigma(X_*^{\text{tgt}}, X_*^{\text{tgt}}) = \mathcal{K}^{DA}\left(X_*^{\text{tgt}}, X_*^{\text{tgt}}\right) + \Theta_{DA}\left(X_*^{\text{tgt}}, X\right) \Theta_{DA}(X, X)^{-1} \mathcal{K}^{DA} \Theta_{DA}(X, X)^{-1} \Theta_{DA}\left(X, X_*^{\text{tgt}}\right)$$
$$- \left(\Theta_{DA}\left(X_*^{\text{tgt}}, X\right) \Theta_{DA}(X, X)^{-1} \mathcal{K}^{DA}\left(X, X_*^{\text{tgt}}\right) + h.c.\right)$$

where "$+h.c.$" means "plus the Hermitian conjugate".

*Proof.* The objective function of Eq. (7) can be rewritten as follows.

$$\mathcal{L}(\theta) = \frac{\alpha}{2n_{\text{src}}} \sum_{i=1}^{n_{\text{src}}} \left( \tilde{f}(x_i^{\text{src}}, \mathbb{P}^{\text{src}}) - y_i^{\text{src}} \right)^2 + \frac{1-\alpha}{2n_{\text{tgt}}^l} \sum_{j=1}^{n_{\text{tgt}}^l} \left( \tilde{f}(x_j^{\text{tgt}}, \mathbb{P}^{\text{tgt}}) - y_j^{\text{tgt}} \right)^2 + \frac{\mu}{2} \hat{\text{MMD}}_{\Theta_{DA}}^2 \left( \mathbb{P}^{\text{src}}, \mathbb{P}^{\text{tgt}} \right)$$

$$= \frac{1}{2} \left\| C\tilde{f}(X) - CY \right\|_2^2 + \frac{\mu}{2} \hat{\text{MMD}}_{\Theta_{DA}}^2 \left( \mathbb{P}^{\text{src}}, \mathbb{P}^{\text{tgt}} \right)$$

where $\tilde{f}(X) = \text{vec}(\{\tilde{f}(x_i)\}_{i=1}^{n_{\text{src}}+n_{\text{tgt}}^l})$, and $C = \text{diag}\{\underbrace{\sqrt{\alpha/n_{\text{src}}}, \cdots, \sqrt{\alpha/n_{\text{src}}}}_{n_{\text{src}}}, \underbrace{\sqrt{(1-\alpha)/n_{\text{tgt}}^l}, \cdots, \sqrt{(1-\alpha)/n_{\text{tgt}}^l}}_{n_{\text{tgt}}^l}\}$ is a constant diagonal matrix.
Then the tangent kernel can be defined as

$$\Theta_{DA}(X, X) = \lim_{d \to \infty} \nabla_{\theta_0} \tilde{f}(X) \nabla_{\theta_0} \tilde{f}(X)^T$$

Moreover, we obtain

$$\nabla_{\theta_0}\tilde{f}(x,\mathbb{P}) = \nabla_{\theta_0}f(x)\cdot g_{w_g}(\mathbb{P}|x)$$

$$\lim_{d\to\infty}\left\langle\nabla_{\theta_0}\tilde{f}(x,\mathbb{P}),\nabla_{\theta_0}\tilde{f}(x',\mathbb{P}')\right\rangle = \Theta(x,x')\cdot\mathcal{K}_{\mathscr{P}|\mathcal{X}}\left(\mathbb{P},\mathbb{P}'|x,x'\right)$$

$$\Theta_{DA}(X,X) = \Theta(X,X)\odot\mathcal{K}_{\mathscr{P}|\mathcal{X}}(X,X)$$

where $\Theta(x,x') = \lim_{d\to\infty}\langle\nabla_{\theta_0}f(x),\nabla_{\theta_0}f(x')\rangle$ is standard neural tangent kernel [31] induced by $f(\cdot)$ with infinite width.

Following [35], we have the linearized neural network given by its first-order Taylor expansion.

$$f_{\theta_t}^{\text{lin}}(x) = f_{\theta_0}(x) + \nabla_{\theta_0}f_{\theta_0}(x)(\theta_t - \theta_0)$$

and $f_{\theta_t}(x) - f_{\theta_t}^{\text{lin}}(x) = \mathcal{O}(1/\sqrt{d^*}) \to 0$ $(d^* \to 0)$. Then we have $\tilde{f}_{\theta_t}(X) = \tilde{f}_{\theta_0}(X) + \nabla_{\theta_0}\tilde{f}_{\theta_0}(X)(\theta_t - \theta_0) + \mathcal{O}(1/\sqrt{d^*})$ where $d^*$ denotes the network width. Using the linearized function $\tilde{f}_{\theta_t}^{\text{lin}}(X) = \tilde{f}_{\theta_0}(X) + \nabla_{\theta_0}\tilde{f}_{\theta_0}(X)(\theta_t - \theta_0)$, we have the dynamics of linearized neural network as follows.

$$\dot{\theta}_t = -\eta\nabla_\theta\mathcal{L}(\theta) = -\eta\nabla_\theta\tilde{f}_{\theta_t}^{\text{lin}}(X)^T C^T C\left(\tilde{f}_{\theta_t}^{\text{lin}}(X) - Y\right)$$

$$= -\eta\nabla_\theta\tilde{f}_{\theta_0}(X)^T C^T C\left(\tilde{f}_{\theta_0}(X) + \nabla_{\theta_0}\tilde{f}_{\theta_0}(X)(\theta_t - \theta_0) - Y\right)$$

Then the ODE has closed form solution as

$$\theta_t = \theta_0 - \nabla_\theta\tilde{f}_{\theta_0}(X)^T\Theta_{DA}^{-1}\left(\mathbb{I} - e^{-\eta\Theta_{DA}C^T Ct}\right)\left(\tilde{f}_{\theta_0}(X) - Y\right)$$

Then given random initialization $\theta_0$, the predictions of this neural network over training $X$ and testing example $X_*^{\text{tgt}}$ are

$$\tilde{f}_{\theta_t}(X) \approx \tilde{f}_{\theta_t}^{\text{lin}}(X) = \tilde{f}_{\theta_0}(X) - \left(\mathbb{I} - e^{-\eta\Theta_{DA}C^T Ct}\right)\left(\tilde{f}_{\theta_0}(X) - Y\right)$$

$$\tilde{f}_{\theta_t}(X_*^{\text{tgt}}) \approx \tilde{f}_{\theta_t}^{\text{lin}}(X_*^{\text{tgt}}) = \tilde{f}_{\theta_0}(X_*^{\text{tgt}}) - \Theta_{DA}(X_*^{\text{tgt}},X)\Theta_{DA}(X,X)^{-1}\left(\mathbb{I} - e^{-\eta\Theta_{DA}C^T Ct}\right)\left(\tilde{f}_{\theta_0}(X) - Y\right)$$

with up to an error of $\mathcal{O}(1/\sqrt{d^*})$.

Here, the minimum eigenvalue $\lambda_{\min}(\Theta_{DA}C^T C) \geq \lambda_{\min}(\Theta_{DA})\lambda_{\min}(C^T C) = \lambda_{\min}(\Theta_{DA}) \cdot \min\{\alpha/n_{\text{src}}, (1-\alpha)/n_{\text{tgt}}^l\}$. Following [27], using $\lambda_{\min}(\Theta(X,X)) > 0$ and $\lambda_{\min}(\mathcal{K}_{\mathscr{P}|\mathcal{X}}(X,X)) > 0$, we have $\lambda_{\min}(\Theta_{DA}) \geq 0$. When $\alpha \in (0,1)$, $\lim_{t\to\infty}\lim_{d^*\to\infty}\tilde{f}_{\theta_t}(X_*^{\text{tgt}}) = \tilde{f}_{\theta_0}(X_*^{\text{tgt}}) - \Theta_{DA}(X_*^{\text{tgt}},X)\Theta_{DA}(X,X)^{-1}\left(\tilde{f}_{\theta_0}(X) - Y\right)$.

Over random initialization of $\theta_0$, $\tilde{f}(X_*^{\text{tgt}})$ converges to a Gaussian distribution, as it is a linear transformation of $\tilde{f}_{\theta_0}(X_*^{\text{tgt}})$ associated with Gaussian distribution. Using $\mathbb{E}[\tilde{f}_{\theta_0}(X_*^{\text{tgt}})] = 0$ and $\mathbb{E}[\tilde{f}_{\theta_0}(X_*^{\text{tgt}})\tilde{f}_{\theta_0}(X_*^{\text{tgt}})] = \mathcal{K}^{DA}(X_*^{\text{tgt}},X_*^{\text{tgt}})$, when $\alpha \in (0,1)$, we have the following results:

$$\mu(X_*^{\text{tgt}}) = \Theta_{DA}(X_*^{\text{tgt}},X)\Theta_{DA}(X,X)^{-1}Y$$

$$\Sigma(X_*^{\text{tgt}},X_*^{\text{tgt}}) = \mathcal{K}^{DA}(X_*^{\text{tgt}},X_*^{\text{tgt}}) + \Theta_{DA}(X_*^{\text{tgt}},X)\Theta_{DA}(X,X)^{-1}\mathcal{K}^{DA}\Theta_{DA}(X,X)^{-1}\Theta_{DA}(X,X_*^{\text{tgt}})$$
$$- \left(\Theta_{DA}(X_*^{\text{tgt}},X)\Theta_{DA}(X,X)^{-1}\mathcal{K}^{DA}(X,X_*^{\text{tgt}}) + h.c.\right)$$

which completes the proof. $\qquad\square$

### A.5   Proof of Theorem 4.5

**Theorem 4.5.** Assume for any training example $(x,y) \in \mathcal{X} \times \mathcal{Y}$, we have $(\tilde{f}(x,\mathbb{P}) - y)^2 \leq M_0$ for some constant $M_0 \geq 0$. Let $\tilde{\mathcal{F}}$ be the hypothesis space induced by infinitely-wide neural networks $\tilde{f}$. Then, for any $\tilde{f} \in \tilde{\mathcal{F}}$ and $\delta > 0$, with probability at least $1 - \delta$, the expected error in the target domain can be bounded as follows.

$$\mathcal{E}_{\mathbb{P}^{\text{tgt}}}(\tilde{f}) \leq \frac{\alpha}{n_{\text{src}}}\sum_{i=1}^{n_{\text{src}}}\left(\tilde{f}(x_i^{\text{src}},\mathbb{P}^{\text{src}}) - y_i^{\text{src}}\right)^2 + \frac{1-\alpha}{n_{\text{tgt}}^l}\sum_{j=1}^{n_{\text{tgt}}^l}\left(\tilde{f}(x_j^{\text{tgt}},\mathbb{P}^{\text{tgt}}) - y_j^{\text{tgt}}\right)^2 + 8\alpha M_0\cdot\text{MMD}_{\Theta_{DA}}\left(\mathbb{P}^{\text{src}},\mathbb{P}^{\text{tgt}}\right)$$

$$+ 2\alpha\Re_{n_{\text{src}}}(\mathcal{H}_{\text{src}}) + 2(1-\alpha)\Re_{n_{\text{tgt}}^l}(\mathcal{H}_{\text{tgt}}) + M\sqrt{\frac{(n_{\text{src}} + n_{\text{tgt}}^l)\log(1/\delta)}{2}}$$

where $\mathcal{H}_{\text{src}} = \{(x,y) \to (\tilde{f}(x^{\text{src}}, \mathbb{P}^{\text{src}}) - y^{\text{src}})^2 : \tilde{f} \in \tilde{\mathcal{F}}\}$ is a set of functions, $\Re_{n_{\text{src}}}(\mathcal{H}_{\text{src}})$ is the Rademacher complexity of $\mathcal{H}_{\text{src}}$ given $n_{\text{src}}$ examples, and $M = \max\{\alpha M_0/n_{\text{src}}, (1-\alpha)M_0/n_{\text{tgt}}^l\}$.

*Proof.* Let $\Psi(X) = \sup_{\tilde{f} \in \tilde{\mathcal{F}}} \mathcal{L}_{\mathbb{P}^{\text{tgt}}}(\tilde{f}) - \alpha \cdot \mathcal{L}_{\hat{\mathbb{P}}^{\text{src}}}(\tilde{f}) - (1-\alpha) \cdot \mathcal{L}_{\hat{\mathbb{P}}^{\text{tgt}}}(\tilde{f})$ where $\mathcal{L}_{\hat{\mathbb{P}}^{\text{src}}}(\tilde{f}) = \frac{1}{n_{\text{src}}}\sum_{i=1}^{n_{\text{src}}} \left(\tilde{f}(x_i^{\text{src}}, \mathbb{P}^{\text{src}}) - y_i^{\text{src}}\right)^2$ and $\mathcal{L}_{\hat{\mathbb{P}}^{\text{tgt}}}(\tilde{f}) = \frac{1}{n_{\text{tgt}}^l}\sum_{j=1}^{n_{\text{tgt}}^l} \left(\tilde{f}(x_j^{\text{tgt}}, \mathbb{P}^{\text{tgt}}) - y_j^{\text{tgt}}\right)^2$.

Then when one point of $X$ is changed, $\Psi(X)$ will change at most $M = \max\{\alpha M_0/n_{\text{src}}, (1-\alpha)M_0/n_{\text{tgt}}^l\}$. Using McDiarmid's inequality, it holds that

$$\Pr[\Psi(X) - \mathbb{E}[\Psi(X)] > \epsilon] \le \exp\left(-\frac{2\epsilon^2}{(n_{\text{src}} + n_{\text{tgt}}^l)M^2}\right)$$

Thus, for any $\delta > 0$, with probability at least $1 - \delta$, we have

$$\mathcal{L}_{\mathbb{P}^{\text{tgt}}}(\tilde{f}) \le \alpha \cdot \mathcal{L}_{\hat{\mathbb{P}}^{\text{src}}}(\tilde{f}) + (1-\alpha) \cdot \mathcal{L}_{\hat{\mathbb{P}}^{\text{tgt}}}(\tilde{f}) + \mathbb{E}[\Psi(X)] + M\sqrt{\frac{(n_{\text{src}} + n_{\text{tgt}}^l)\log(1/\delta)}{2}}$$

Moreover,

$$\mathbb{E}[\Psi(X)] = \mathbb{E}\left[\sup_{\tilde{f} \in \tilde{\mathcal{F}}} \alpha \cdot \mathcal{L}_{\hat{\mathbb{P}}^{\text{src}}}(\tilde{f}) + (1-\alpha) \cdot \mathcal{L}_{\hat{\mathbb{P}}^{\text{tgt}}}(\tilde{f}) - \mathcal{L}_{\mathbb{P}^{\text{tgt}}}(\tilde{f})\right]$$

$$\le \alpha \cdot \mathbb{E}\left[\sup_{\tilde{f} \in \tilde{\mathcal{F}}} \mathcal{L}_{\hat{\mathbb{P}}^{\text{src}}}(\tilde{f}) - \mathcal{L}_{\mathbb{P}^{\text{tgt}}}(\tilde{f})\right] + (1-\alpha) \cdot \mathbb{E}\left[\sup_{\tilde{f} \in \tilde{\mathcal{F}}} \mathcal{L}_{\hat{\mathbb{P}}^{\text{tgt}}}(\tilde{f}) - \mathcal{L}_{\mathbb{P}^{\text{tgt}}}(\tilde{f})\right]$$

$$\le 2\alpha\Re_{n_{\text{src}}}(\mathcal{H}_{\text{src}}) + 2(1-\alpha)\Re_{n_{\text{tgt}}^l}(\mathcal{H}_{\text{tgt}}) + \alpha \cdot \sup_{\tilde{f} \in \tilde{\mathcal{F}}} \mathcal{L}_{\mathbb{P}^{\text{src}}}(\tilde{f}) - \mathcal{L}_{\mathbb{P}^{\text{tgt}}}(\tilde{f})$$

When $(\tilde{f}(x,\mathbb{P}) - y)_2 \le M_0$ for any training example $(x,y) \in \mathcal{X} \times \mathcal{Y}$, Lemma 23 of [13] shows that $\left|(\tilde{f}(x,\mathbb{P}) - y)^2 - (\tilde{f}'(x,\mathbb{P}) - y)^2\right| \le 2M_0\left|\tilde{f}(x,\mathbb{P}) - \tilde{f}'(x,\mathbb{P})\right|$. Therefore, we have $(\tilde{f}(x,\mathbb{P}) - y)^2 \le 2M_0\left|\tilde{f}(x,\mathbb{P}) - y\right|$. It is shown in [31] that a infinitely-wide neural network would behave as its linearization around the initialization and achieves zero training loss under MSE loss. Thus, there exists a small perturbation $\Delta\theta$ over parameters for each example $(x,y)$ such that $\tilde{f}_{\theta+\Delta\theta}(x,\mathbb{P}) = y$.

$$\sup_{\tilde{f} \in \tilde{\mathcal{F}}} \mathcal{L}_{\mathbb{P}^{\text{src}}}(\tilde{f}) - \mathcal{L}_{\mathbb{P}^{\text{tgt}}}(\tilde{f}) \le \sup_{\tilde{f} \in \tilde{\mathcal{F}}} \left|\mathbb{E}_{\mathbb{P}^{\text{src}}}\left(\tilde{f}(x^{\text{src}}, \mathbb{P}^{\text{src}}) - y^{\text{src}}\right)^2 - \mathbb{E}_{\mathbb{P}^{\text{tgt}}}\left(\tilde{f}(x^{\text{tgt}}, \mathbb{P}^{\text{tgt}}) - y^{\text{tgt}}\right)^2\right|$$

$$\le 4M_0 \cdot \sup_{\tilde{f} \in \tilde{\mathcal{F}}} \left|\mathbb{E}_{\mathbb{P}^{\text{src}}}\left|\tilde{f}(x^{\text{src}}, \mathbb{P}^{\text{src}}) - y^{\text{src}}\right| - \mathbb{E}_{\mathbb{P}^{\text{tgt}}}\left|\tilde{f}(x^{\text{tgt}}, \mathbb{P}^{\text{tgt}}) - y^{\text{tgt}}\right|\right|$$

$$\le 8M_0 \cdot \sup_{\tilde{f} \in \tilde{\mathcal{F}}} \left|\mathbb{E}_{\mathbb{P}^{\text{src}}}\left[\tilde{f}(x^{\text{src}}, \mathbb{P}^{\text{src}}) - y^{\text{src}}\right] - \mathbb{E}_{\mathbb{P}^{\text{tgt}}}\left[\tilde{f}(x^{\text{tgt}}, \mathbb{P}^{\text{tgt}}) - y^{\text{tgt}}\right]\right|$$

$$\le 8M_0 \cdot \sup_{\Delta\theta \in \mathcal{H}_{DA}, \tilde{f} \in \tilde{\mathcal{F}}} \left|\mathbb{E}_{\mathbb{P}^{\text{src}}}\left[\tilde{f}_\theta(x^{\text{src}}, \mathbb{P}^{\text{src}}) - \tilde{f}_{\theta+\Delta\theta}(x^{\text{src}}, \mathbb{P}^{\text{src}})\right] - \mathbb{E}_{\mathbb{P}^{\text{tgt}}}\left[\tilde{f}(x^{\text{tgt}}, \mathbb{P}^{\text{tgt}}) - \tilde{f}_{\theta+\Delta\theta}(x^{\text{tgt}}, \mathbb{P}^{\text{tgt}})\right]\right|$$

$$= 8M_0 \cdot \sup_{\Delta\theta \in \mathcal{H}_{DA}, \tilde{f} \in \tilde{\mathcal{F}}} \left|\mathbb{E}_{\mathbb{P}^{\text{src}}}\left[\nabla_\theta \tilde{f}_\theta(x^{\text{src}}, \mathbb{P}^{\text{src}})\Delta\theta\right] - \mathbb{E}_{\mathbb{P}^{\text{tgt}}}\left[\nabla_\theta \tilde{f}(x^{\text{tgt}}, \mathbb{P}^{\text{tgt}})\Delta\theta\right]\right|$$

$$= 8M_0 \cdot \text{MMD}_{\Theta_{DA}}\left(\mathbb{P}^{\text{src}}, \mathbb{P}^{\text{tgt}}\right)$$

which $\mathcal{H}_{DA}$ is the RKHS induced by kernel $\Theta_{DA}$. □

### A.6 Proof of Corollary 4.6

**Corollary 4.6.** With the assumptions in Lemma 4.1, when $\beta_{x,\tilde{x}_i} = 1/\|\sum_{i=1}^n \langle\cdot, \tilde{x}_i\rangle_{\mathcal{K}_{\mathcal{X}}}\|$, the Gaussian process induced by $\tilde{f}(x)$ at the initialization would be equivalent to the adaptive Gaussian process in [7] over NNKP kernel.

*Proof.* One special case of distribution-informed neural network at initialization is that when the coefficient $\beta_{x,\tilde{x}_i} = 1/||\sum_{i=1}^n \langle \cdot, \tilde{x}_i \rangle_{\mathcal{K}_\mathcal{X}}||$, it can be seen that $\Phi_x(\mathbb{P}) = \frac{1}{||\sum_{i=1}^n \langle \cdot, \tilde{x}_i \rangle_{\mathcal{K}_\mathcal{X}}||} \sum_{i=1}^n \langle \cdot, \tilde{x}_i \rangle_{\mathcal{K}_\mathcal{X}}$ is the normalized mean mapping [24] of data distribution $\mathbb{P}$ in the NNGP kernel space. In this case, the distribution kernel $\mathcal{K}_{\mathscr{P}|\mathcal{X}}(\mathbb{P}, \mathbb{P}'|x, x')$ in Eq. (6) can be explained as the inner product of the mean mappings of $\mathbb{P}$ and $\mathbb{P}'$. Moreover, using the definition of maximum mean discrepancy (MMD) [24], it can be shown that $\mathcal{K}_{\mathscr{P}|\mathcal{X}}(\mathbb{P}, \mathbb{P}'|x, x') = \frac{c - \hat{\text{MMD}}^2_{\mathcal{K}_\mathcal{X}}(\mathbb{P}, \mathbb{P}')}{\Phi_x(\mathbb{P})\Phi_x(\mathbb{P}')}$, where $\hat{\text{MMD}}_{\mathcal{K}_\mathcal{X}}(\cdot)$ denotes the empirical MMD in the NNGP kernel space, and $c = \frac{1}{n^2} \sum_{i,j=1}^n \mathcal{K}_\mathcal{X}(\tilde{x}_i, \tilde{x}_j) + \frac{1}{n'^2} \sum_{i,j=1}^{n'} \mathcal{K}_\mathcal{X}(\tilde{x}'_i, \tilde{x}'_j)$. It implies that $\mathcal{K}_{\mathscr{P}|\mathcal{X}}(\mathbb{P}, \mathbb{P}'|x, x')$ is negatively correlated with the popular MMD estimator. Moreover, when $\mathbb{P} = \mathbb{P}'$, $\mathcal{K}_{\mathscr{P}|\mathcal{X}}(\mathbb{P}, \mathbb{P}'|x, x') = 1$, otherwise $\mathcal{K}_{\mathscr{P}|\mathcal{X}}(\mathbb{P}, \mathbb{P}'|x, x') = \frac{1}{||\sum_{i=1}^n \langle \cdot, \tilde{x}_i \rangle_{\mathcal{K}_\mathcal{X}}|| \cdot ||\sum_{j=1}^{n'} \langle \cdot, \tilde{x}'_j \rangle_{\mathcal{K}_\mathcal{X}}||} \sum_{i=1}^n \sum_{j=1}^{n'} \mathcal{K}_\mathcal{X}(\tilde{x}_i, \tilde{x}'_j)$. In this case, the adaptive transfer kernel $\mathcal{K}^{DA}(X, X)$ induced by $\tilde{f}(x)$ at the initialization would be equivalent to [7], by setting the transfer parameter $\tau$ of Eq. (3) as $\mathcal{K}_{\mathscr{P}|\mathcal{X}}(\mathbb{P}^{\text{src}}, \mathbb{P}^{\text{tgt}}|x^{\text{src}}, x^{\text{tgt}})$ for any inputs $x^{\text{src}}, x^{\text{tgt}}$. $\square$

## A.7 Proof of Theorem A.1

**Theorem A.1.** When $w_i^{\text{src}} > 0$ for all $i = 1, \cdots, n_{\text{src}}$, the reweighting domain adaptation approach Eq. (9) and standard supervised learning have identical predictions on test target data, i.e., $\lim_{t\to\infty} \lim_{d^*\to\infty} f_{\theta_t}(X_*^{\text{tgt}}) = f_{\theta_0}(X_*^{\text{tgt}}) - \Theta(X_*^{\text{tgt}}, X)\Theta^{-1}(f_{\theta_0}(X) - Y)$.

*Proof.* The objective function of Eq. (7) can be rewritten as follows.

$$\mathcal{L}_{RW}(\theta) = \frac{\alpha}{2n_{\text{src}}} \sum_{i=1}^{n_{\text{src}}} w_i^{\text{src}} ||f(x_i^{\text{src}}) - y_i^{\text{src}}||_2^2 + \frac{1-\alpha}{2n_{\text{tgt}}} \sum_{j=1}^{n_{\text{tgt}}^l} ||f(x_j^{\text{tgt}}) - y_j^{\text{tgt}}||_2^2 = \frac{1}{2} ||CBf(X) - CBY||_2^2$$

where $f(X) = \text{vec}(\{f(x_i)\}_{i=1}^{n_{\text{src}}+n_{\text{tgt}}^l})$, and $C = \text{diag}\{\underbrace{\sqrt{\alpha/n_{\text{src}}}, \cdots, \sqrt{\alpha/n_{\text{src}}}}_{n_{\text{src}}}, \underbrace{\sqrt{(1-\alpha)/n_{\text{tgt}}^l}, \cdots, \sqrt{(1-\alpha)/n_{\text{tgt}}^l}}_{n_{\text{tgt}}^l}\}$

is a diagonal matrix and $B = \text{diag}\{\underbrace{\sqrt{w_1^{\text{src}}}, \cdots, \sqrt{w_{n_{\text{src}}}^{\text{src}}}}_{n_{\text{src}}}, \underbrace{1, \cdots, 1}_{n_{\text{tgt}}^l}\}$. We have the dynamics of linearized neural network as follows (let $A = CB$ for brevity).

$$\dot{\theta}_t = -\eta \nabla_\theta \mathcal{L}_2(\theta) = -\eta \nabla_\theta f_{\theta_t}^{\text{lin}}(X)^T A^T A \left(f_{\theta_t}^{\text{lin}}(X) - Y\right)$$
$$= -\eta \nabla_\theta f_{\theta_0}(X)^T A^T A \left(f_{\theta_0}(X) + \nabla_{\theta_0} f_{\theta_0}(X)(\theta_t - \theta_0) - Y\right)$$

Then the ODE has closed form solution as

$$\theta_t = \theta_0 - \nabla_\theta f_{\theta_0}(X)^T \Theta^{-1} \left(\mathbb{I} - e^{-\eta \Theta A^T A t}\right) (f_{\theta_0}(X) - Y)$$

Then given random initialization $\theta_0$, the predictions of this neural network over training $X$ and testing examples $X_*^{\text{tgt}}$ are

$$f_{\theta_t}(X_*^{\text{tgt}}) = f_{\theta_0}(X_*^{\text{tgt}}) - \Theta(X_*^{\text{tgt}}, X)\Theta^{-1} \left(\mathbb{I} - e^{-\eta \Theta A^T A t}\right) (f_{\theta_0}(X) - Y)$$

Therefore, when $w_i^{\text{src}} > 0$ for all $i = 1, \cdots, n_{\text{src}}$, it can be shown that $\lambda_{\min}(\Theta A^T A) \geq \lambda_{\min}(\Theta)\lambda_{\min}(A^T A) > 0$. Then we have $\lim_{t\to\infty} \lim_{d^*\to\infty} f_{\theta_t}(X_*^{\text{tgt}}) = f_{\theta_0}(X_*^{\text{tgt}}) - \Theta(X_*^{\text{tgt}}, X)\Theta^{-1}(f_{\theta_0}(X) - Y)$.

As indicated in previous work [35], the neural network $f(\cdot)$ with standard MSE loss also has the following objective function

$$\mathcal{L}_{sup}(\theta) = \frac{1}{2} \sum_{i=1}^{n_{\text{src}}} ||f(x_i^{\text{src}}) - y_i^{\text{src}}||_2^2 + \frac{1}{2} \sum_{j=1}^{n_{\text{tgt}}^l} ||f(x_j^{\text{tgt}}) - y_j^{\text{tgt}}||_2^2 = \frac{1}{2} ||f(X) - Y||_2^2$$

For any test examples $X_*^{\text{tgt}}$, it has

$$f_{\theta_t}^{sup}(X_*^{\text{tgt}}) = f_{\theta_0}(X_*^{\text{tgt}}) - \Theta(X_*^{\text{tgt}}, X)\Theta^{-1}\left(\mathbb{I} - e^{-\eta\Theta t}\right)(f_{\theta_0}(X) - Y)$$

and $\lim_{t\to\infty}\lim_{d^*\to\infty} f_{\theta_t}^{sup}(X_*^{\text{tgt}}) = f_{\theta_0}(X_*^{\text{tgt}}) - \Theta(X_*^{\text{tgt}}, X)\Theta^{-1}(f_{\theta_0}(X) - Y)$. This indicates that when $w_i^{\text{src}} > 0$ for all $i = 1, \cdots, n_{\text{src}}$, the reweighting domain adaptation approach and standard supervised learning have identical predictions on test target data. $\square$

## A.8 Proof of Corollary 4.7

**Corollary 4.7.** With the assumption in Theorem 4.4, our framework Eq. (7) with distribution-informed neural network $\tilde{f}(\cdot)$ can recover the popular reweighting domain adaptation approach Eq. (9) in the function space.

*Proof.* We can consider the following special case of the distribution representation in Eq. (4). For source example $x$, we can set $\beta_{x,x_i} = \frac{1}{n_{\text{src}}||x_i||_{\mathcal{K}_{\mathcal{X}}}} \cdot \delta[w_i^{\text{src}} > 0]$, where $\delta[\cdot]$ is a Kronecker delta function. For target example $x$, we can set $\beta_{x,x_j} = \frac{1}{n_{\text{tgt}}||x_j||_{\mathcal{K}_{\mathcal{X}}}}$ where $n_{\text{tgt}} = n_{\text{tgt}}^l + n_{\text{tgt}}^u$, as we use all the target training examples as the basis examples for learning the input-oriented representation of the target domain. In this case, for distribution-informed NTK $\Theta_{DA}(X, X)$, we have $\Theta_{DA}(x_i, x_j) = 0$ if $x_i$ (or $x_j$) are source example with $w_i^{\text{src}} > 0$ (or $w_j^{\text{src}} > 0$), $\Theta_{DA}(x_i, x_j) = \Theta(x_i, x_j)$ otherwise. That is, when $w_i^{\text{src}} > 0$ for all $i = 1, \cdots, n_{\text{src}}$, we have $\Theta_{DA}(X, X) = \Theta(X, X)$. Theorem 4.4 indicates that it can produce the same prediction function over random initialization as the reweighting domain adaptation approach Eq. (9). When there are some source examples with $w_i^{\text{src}} = 0$, the objective function of our domain adaptation will also simply filter out those source examples, and the final prediction function depends only on the source examples with $w_i^{\text{src}} > 0$. $\square$

## A.9 Proof of Corollary 4.8

**Corollary 4.8.** Under mild conditions, our framework Eq. (7) with distribution-informed neural network $\tilde{f}(\cdot)$ can recover the standard domain invariant representation learning in Eq. (1), where the domain discrepancy measure $d(\cdot, \cdot)$ is instantiated with MMD in RKHS induced by standard NTK $\Theta$.

*Proof.* It can be shown when the distribution representation of Eq. (4) is shared by all the input examples. In this case, for any $w_g$, $\tilde{f}(x, \mathbb{P}) = a \cdot f_\theta(x)$ where $a = \Phi_x(\mathbb{P})^T w_g \in \mathbb{R}$ is shared by all examples. Then, we have $\tilde{f}(x, \mathbb{P}) = a \cdot f_\theta(x) = a \cdot \phi_{\theta<L}(x)^T w = \phi_{\theta<L}(x)^T(aw)$. Therefore, in this case, $\tilde{f}(\cdot)$ is equivalent to a simple distribution-free neural network. The overall framework Eq. (7) would be degenerated into the standard domain invariant representation learning in Eq. (1). Notice that when using the distribution-free neural network, the framework Eq. (1) requires an additional discrepancy minimization regularization to guarantee the success of domain adaptation. In our case, the discrepancy minimization regularization would be given by the MMD in RKHS induced by the NTK $\Theta_{DA}$. Here, when the distribution representation of Eq. (4) is shared by all the input examples, it holds that for any $x, x' \in \mathcal{X}$, $\Theta_{DA}(x, x') = b \cdot \Theta(x, x')$ where $b = ||\Phi_x(\mathbb{P})||_{\mathcal{K}_{\mathcal{X}}}^2 \in \mathbb{R}$. Thus, $\hat{\text{MMD}}_{\Theta_{DA}}^2(\mathbb{P}^{\text{src}}, \mathbb{P}^{\text{tgt}}) = b \cdot \hat{\text{MMD}}_\Theta^2(\mathbb{P}^{\text{src}}, \mathbb{P}^{\text{tgt}})$. $\square$

## A.10 Algorithms

**DINO-INIT vs. DINO-TRAIN:** `DINO-INIT` is weakly-trained (i.e., only the last layer is trained), while `DINO-TRAIN` is fully-trained (i.e., all network layers are trained). We observe from the experimental results that the weakly-trained `DINO-INIT` might outperform `DINO-TRAIN` in some cases. This observation is consistent with previous work [33] on standard neural networks.

The overall training procedures of `DINO-INIT` are illustrated in Algorithm 1. Here we use both labeled target examples and unlabeled target examples $X_l^{\text{tgt}} \cup X_u^{\text{tgt}}$ as the basis target examples of Eq. (4). This allows the proposed `DINO-INIT` to be applied for domain adaptation scenarios [5, 13, 47] where both limited labeled and adequate unlabeled target data are available during model training. When no unlabeled target data is available [7], i.e., only limited labeled target data is given, we can simply use those labeled target examples as the basis target examples of Eq. (4).

---

**Algorithm 1** `DINO` Algorithms

---

**Input:** Labeled source examples $(X^{\text{src}}, Y^{\text{src}})$, labeled target examples $(X_l^{\text{tgt}}, Y_l^{\text{tgt}})$ and unlabeled target examples $X_u^{\text{tgt}}$, neural network architecture of $f(\cdot)$.

**Output:** Predictions on testing target examples $X_*^{\text{tgt}}$.

1: Set $X^{\text{src}}$ to be basis source examples of Eq. (4);
2: Set $X_l^{\text{tgt}} \cup X_u^{\text{tgt}}$ to be basis target examples of Eq. (4);
3: Calculate the basis NNGP kernels of Eq. (6);
4: Estimate the coefficient $w^r$ and noise variance $\sigma_{\text{src}}$, $\sigma_{\text{tgt}}$ by maximizing $p(Y_l^{\text{tgt}}|X_l^{\text{tgt}}, X^{\text{src}}, Y^{\text{src}})$;
5: Calculate the posterior distribution with $\bar{\mu}$ and $\bar{\Sigma}$;
6: Output $\hat{Y}_*^{\text{tgt}}|X_*^{\text{tgt}} \sim \mathcal{N}(\bar{\mu}, \bar{\Sigma})$.

---

In addition, the overall training procedures of `DINO-TRAIN` are illustrated in Algorithm 2. Note that in this paper, we focus on analyzing the training dynamics of our model in the adaptation scenarios where limited labeled data in the target domain is available. But our theoretical results can be extended to unsupervised domain adaptation setting where only unlabeled data is available in the target domain. For example, the proposed `DINO-TRAIN` framework of Eq. (7) can be trained with $\alpha = 1$[2] for unsupervised domain adaptation. Then we can show similar convergence and generalization results of this framework.

**Analysis of the reweighting approach Eq. (9):** Notice that if we do not consider the distribution shift between source and target domain, the prediction function of reweighting approach Eq. (9) can be simply learned by minimizing the standard supervised learning loss: $\mathcal{L}_{sup}(\theta) = \frac{1}{2}\sum_{i=1}^{n_{\text{src}}} \left(f(x_i^{\text{src}}) - y_i^{\text{src}}\right)_2^2 + \frac{1}{2}\sum_{j=1}^{n_{\text{tgt}}^l} \left(f(x_j^{\text{tgt}}) - y_j^{\text{tgt}}\right)_2^2$ over all the labeled training examples. The following theorem shows that for an infinitely-wide neural network $f(\cdot)$, the reweighting domain adaptation approach Eq. (9) and the standard supervised learning have identical prediction function when they converges.

**Theorem A.1.** *When $w_i^{src} > 0$ for all $i = 1, \cdots, n_{src}$, in the limit of infinite network width, the reweighting domain adaptation approach Eq. (9) and the standard supervised learning would have identical prediction function on testing target data, i.e., $\lim_{t\to\infty} f_{\theta_t}(X_*^{tgt}) = f_{\theta_0}(X_*^{tgt}) - \Theta(X_*^{tgt}, X)\Theta^{-1}(f_{\theta_0}(X) - Y)$.*

This result can be generalized to the scenarios where $w_i^{\text{src}} \geq 0$. In this case, the reweighting domain adaptation approach Eq. (9) considers only the source examples with $w_i^{\text{src}} > 0$. It is equivalent to the standard supervised learning over those source examples. Therefore, we have the following observations. (1) The weight $w_i^{\text{src}}$ can only filter out some unrelated source examples (i.e., $w_i^{\text{src}} = 0$) for the reweighting approach. (2) For neural networks with infinite width, the learned predictive functions of the reweighting approach and the standard supervised learning are equivalent in the function space. Compared to the reweighting approach Eq. (9), our framework of Eq. (7) learns the distribution-informed representation for both source and target data. The resulting prediction function is explicitly determined by the domain discrepancy (indicated by the distribution-informed NTK $\Theta_{DA}$).

**Extension:** We would like to point out that the proposed DINO framework can be easily generalized to other network architectures. This is because previous works [4, 22, 54, 55] have shown the existence of NNGP and NTK in different network architectures, including (residual) convolutional neural networks, recurrent neural networks, transformer, etc. Therefore, we can adapt our algorithms and theoretical analysis to those network architectures as well. In this paper, we focus on the most used fully connected network, and the exploration of network architecture comparison in our framework is beyond the scope of this paper.

**Limitations:** In this paper, we assume that there is only one source domain. But in real scenarios, it is possible to gather source information from multiple domains. In the context of multi-source domain adaptation regression, the generalization and convergence analysis of domain adaptation regression with neural network might provide the insight on selecting high-quality source data for better knowledge transfer. We would like to leave it as our future work.

---

[2]In Subsection 4.3, we consider $\alpha \in (0, 1)$, as the framework with $\alpha = 1$ or $\alpha = 0$ will produce a slightly different convergence result.

---

**Algorithm 2** `DINO-TRAIN`

---

**Input:** Labeled source examples $(X^{\text{src}}, Y^{\text{src}})$, labeled target examples $(X_l^{\text{tgt}}, Y_l^{\text{tgt}})$ and unlabeled target examples $X_u^{\text{tgt}}$, neural network architecture of $f(\cdot)$.

**Output:** Predictions on testing target examples $X_*^{\text{tgt}}$.

1: Set $X^{\text{src}}$ to be basis source examples of Eq. (4);
2: Set $X_l^{\text{tgt}} \cup X_u^{\text{tgt}}$ to be basis target examples of Eq. (4);
3: Calculate the basis NNGP kernels of Eq. (6);
4: **repeat**
5:     Minimize the objective function of Eq. (7);
6: **until** It is converged
7: Output $\hat{Y}_*^{\text{tgt}} = \tilde{f}(X_*^{\text{tgt}})$.

---

### A.11 Experimental Details

#### A.11.1 Data Sets

**dSprites** [40]: It is composed of 737,280 images from three domains: Color (C), Noisy (N) and Scream (S). Following [10], we evaluate all the baselines on six adaptation benchmarks: C → N, C → S, N → C, N → S, S → C, and S → N. For each image, it has three regression tasks, i.e., predicting three factors of variations (scale, position X and Y).

**MPI3D** [23]: It contains over 3M images from three domains: Toy (T), Realistic (RC) and Real (RL). We evaluate all the baselines on six benchmarks: T → RC, T → RL, RC → T, RC → RL, RL → T, and RL → RC. It is shown [10] that for each image, it involves two regression tasks, i.e., predicting three factors of variations (position X and Y).

**Plant Phenotyping**: It aims to predict diverse traits (e.g., Nitrogen) of plants related to the plants' growth using leaf hyperspectral reflectance (i.e., spectral wavelengths 500-2400 nm). Then the input example with spectral wavelengths 500-2400 nm is formulated as a 1901-dimensional feature vector. Here we consider the following two domains [46]: Maize (M) and Maize_UNL (MU), where "M" denotes the collected maize data from Illinois and "MU" denotes the collected maize data from Nebraska. In our case, the task is to predict the Nitrogen content of maize using the leaf hyperspectral reflectance.

For dSprites and MPI3D, following [10], we use the default train/test split for the target domain. In this case, we randomly choose 100 training target images as the labeled examples, and others as the unlabeled ones for all our experiments. For Plant Phenotyping, for the target domain, we randomly 5% of data as the label examples, and others as the unlabeled ones.

#### A.11.2 Implementations

All the experiments are performed on a Windows machine with four 3.80GHz Intel Cores, 64GB RAM and two NVIDIA Quadro RTX 5000 GPUs. In the experiments, our algorithms are implemented based on a $L$-layer ($L = 6$) fully-connected neural network with ReLU activation function. In dSprites and MPI3D data sets, the size of input images is $64 \times 64 \times 3$, we simply vectorize the input image to a 12288-dimensional vector. That is, each input image $x \in \mathbb{R}^{12288}$ can be learned by the fully-connected neural network. The NNGP and neural tangent kernels induced by this neural network can be estimated using the Neural Tangents package [41]. In addition, we set $\alpha = 0.5$ and $\mu = 0.1$ for our `DINO-TRAIN` method. Note that different from previous works [37, 10], all the baselines will be trained from scratch and use the same neural network architecture for domain adaptation. It is shown [60, 10] that initializing the neural network using existing pre-trained models (e.g. ResNet-50 [26]) can improve the domain adaptation performance, but it is out of the scope of this paper.

In our experiments, we use three deep domain adaptation baselines: DAN [37], WANN [16] and RSD [10]. DAN [37] and RSD [10] focus on domain invariant representation learning (see Subsection 3.2.1) by empirically minimizing the prediction error over the labeled training examples and the domain discrepancy (i.e., maximum mean discrepancy or representation subspace distance). WANN uses the reweighting technique in Subsection 3.2.2. Notice that DAN and RSD are proposed for unsupervised domain adaptation with no label information from the target domain. In the experiments,

for a fair comparison, we extend them to domain adaptation scenarios with little label information from the target domain, by adding the prediction error over the labeled training target examples.

Besides, Figure 3b shows the comparison of the proposed gradient-based MMD $\hat{\text{MMD}}_{\Theta_{DA}} (\cdot, \cdot)$ and conventional MMD with RBF kernel for domain adaptation in Subsection 5.2. To be more specific, we consider a simple 3-layer fully-connected neural network with ReLU. Following [37], we implement the domain adaptation approach by minimizing the prediction error over the labeled training examples and the MMD with RBF kernel over the source and target features learned by the first $l = 1, 2, 3$ layers. We denote those approaches as 'MMD-RBF (layer 1)', 'MMD-RBF (layer 2)' and 'MMD-RBF (layer 3)' in Figure 3b, respectively. For a fair comparison, we use the same neural network architecture to implement the domain adaptation approach with our proposed $\hat{\text{MMD}}_{\Theta_{DA}} (\cdot, \cdot)$, which is denoted as 'MMD-NTK' in Figure 3b.