# OpenReview forum: "Distribution-Informed Neural Networks for Domain Adaptation Regression"
_NeurIPS.cc/2022/Conference — NeurIPS 2022 Accept_

### Official Review · Reviewer_2Ab3 · 2022-07-10

**Rating:** 6
**Confidence:** 2
**Soundness:** 3 good
**Presentation:** 3 good
**Contribution:** 3 good

**Summary:**

The paper proposes to use a distribution-informed neural network for domain adaptation regression. The input-oriented feature representation is defined based on its distance to other examples in the same domain, thereby achieving distribution-conditioned representation of the input data. Theoretical analysis shows that the proposed distribution-informed neural network is equivalent to domain adaptive Gaussian Process.
Empirical results on dSprites and MPI3D demonstrate that the proposed approach outperforms Gaussian Process, domain adaptive Gaussian Process and deep domain adaptation methods.

**Questions:**

Q1-2: Please refer to my question in the weakness section.

Q3. In Eq. 5, it is not clear how $f_\theta(x)=\phi_{\theta<L}(x)^{T} w$ could represent $P(x)$. The probabilistic formation is not enforced in the training process.

**Strengths And Weaknesses:**

Strengths:

1. I think the proposal of distribution-informed neural network that is aware of the input-output training dynamic is novel. Although not mentioned in the paper, the proposed approach is somewhat related to self-attention where the model encodes each individual input based on its relations with other inputs. The illustration in Fig. 1 on how "local model evolution" relates to domain adaptation is insightful.
2. The proposed approach is theoretically grounded and the paper presents a unified framework for analyzing domain discrepancy that is connected to adaptive Gaussian process, sample re-weighting and domain-invariant representation learning.
3. The model outperforms other baselines in the empirical studies.


Weakness:
1. It is unclear what the "training dynamics" is referring to in the proposal. In the introduction, the dynamics are hinting at how the model changes during training, but the main methodological contribution seems to be the "distribution-informed neural network." If the domain divergence is defined on the training dynamics of a model, it would be susceptive to common problems when training a model, e.g., underfitting, overfitting, poor-calibration.
2. Regarding the empirical results on MPI3D, it is counter-intuitive that the weakly supervised version "DINO-INIT" outperforms "DINO-TRAIN". The authors should clarify the difference between the two models and explain why additional training is detrimental.

---

> ### Author Response · Authors · 2022-08-02
> **Response to Reviewer 2Ab3**
>
> Q1: It is unclear what the "training dynamics" is referring to in the proposal. In the introduction, the dynamics are hinting at how the model changes during training, but the main methodological contribution seems to be the "distribution-informed neural network." If the domain divergence is defined on the training dynamics of a model, it would be susceptive to common problems when training a model, e.g., underfitting, overfitting, poor-calibration.
>
> A1: As explained in Lines 67-69, “training dynamics” indicate how the distribution-informed neural network is changed during model training with gradient descent. By analyzing the “training dynamics” of our proposed distribution-informed neural network, we can not only measure the distribution shift of source and target domains (see Eq. (8)), but also show the convergence and generalization results (see Theorems 4.4&4.5) of the proposed algorithm DINO-TRAIN based on the distribution-informed neural network.
>
> We see that all domain adaptation approaches based on neural networks might suffer from common problems, e.g., underfitting, overfitting, poor-calibration. For example, previous work [14] defined the domain discrepancy in the latent feature space learned by a deep neural network. But both our experimental results and previous empirical evaluations [14] show that the domain discrepancy using neural networks could work well for domain adaptation regression in practice. Moreover, we have illustrated in Lines 268-276 that compared to previous work, our discrepancy measure using “training dynamics” can not only unify the domain adaptation regression framework with neural networks, but also enable the theoretical convergence and generalization analysis.
>
> Q2: Regarding the empirical results on MPI3D, it is counter-intuitive that the weakly supervised version "DINO-INIT" outperforms "DINO-TRAIN". The authors should clarify the difference between the two models and explain why additional training is detrimental.
>
> A2: (1) In the revised paper, we add more descriptions of our models, e.g., Lines 212-214.
>
> (2) As explained in Appendix A.10, it is empirically observed [41] that for standard neural networks, NNGP might outperform NTK in some cases. Their performance varies with different model architectures and data sets. One explanation in previous work [32] is that the covariance of Gaussian process induced by NTK is much larger than that induced by NNGP, i.e., $\Sigma_{NTKGP} \succeq \Sigma_{NNGP}$ (or $\Sigma_{NTKGP} - \Sigma_{NNGP}$ is a positive definite kernel). This might lead to an unstable prediction function of Gaussian process induced by NTK. We did have similar observations for the proposed distribution-informed neural network, e.g., DINO-INIT over adaptive NNGP and DINO-TRAIN over distribution-informed NTK. We have added the analysis in Lines 368-370 in the revised paper.
>
> Q3: In Eq. 5, it is not clear how $f_{\theta}(x)=\phi_{\theta^{<L}}^Tw$ could represent $\mathbb{P}(x)$. The probabilistic formation is not enforced in the training process.
>
> A3: We would like to point out that this is a misunderstanding. In Eq. (5), we use $f_{\theta}(x)=\phi_{\theta^{<L}}^Tw$ to denote the feature representation learning of input $x$, and $g_{w_g} (\mathbb{P}|x)=\Phi_x (\mathbb{P})^T w_g$ to denote the input-oriented distribution feature representation of distribution $\mathbb{P}$. As shown in Eq. (4), we learn the input-oriented distribution feature representation $\Phi_x (\mathbb{P})$, which maps the distribution $\mathbb{P}$ into a feature representation in the NNGP kernel space. In the training process, we focus on the empirical distribution with finite training examples. The key idea is to map those training examples into the kernel space and take the weighted average over the feature maps. Thus, no probabilistic formation is involved to learn the input-oriented distribution feature representation of distribution $\mathbb{P}$ during model training. In Lines 197-198, we explain that it degenerates into the mean mapping of $\mathbb{P}$ in the kernel space [31] under mild conditions.

---

> > ### Comment · Reviewer_2Ab3 · 2022-08-08
> > **Thanks for the response**
> >
> > Thanks for the response, all my questions have been answered.

---

### Official Review · Reviewer_WMY4 · 2022-07-11

**Rating:** 5
**Confidence:** 4
**Soundness:** 2 fair
**Presentation:** 3 good
**Contribution:** 2 fair

**Summary:**

In this paper, the authors propose a distribution-informed neural network for domain adaptation regression, called DINO.  The proposed distribution-informed neural network can be explained by existing domain adaptative Gaussian process. The authors also analyze the convergence and generalization bound of DINO. Experiments are done to show the effectiveness of the proposed DINO framework.

**Questions:**

1.The paper is specifically written for domain adaptation regression. However, to this reviewer, the technical contribution is not specific to regression. The authors claim that classification is naturally formulated as regression. Thus, the reviewer wonders how this method performs in those classification tasks (there are quite a lot SOTA UDA papers for classification, the authors may want to compare with some recent ones.).

2.To this reviewer, the related work section should be re-organized and further improved. Compared with discussing the big picture of domain adaptation, the authors should directly focus on the regression works on domain adaptation or transfer learning. The authors may want to add two more paragraphs, one discussing domain adaptation regression works and another one discussing the GP based adaptation setting, as these two research lines are closely related to the current paper.

More importantly, it is more convincing to discuss how those closely related works, e.g., [14] and [11], differ from the current paper, instead of just listing them out.

3.To this reviewer, the setting studied in this paper is different from the settings of [14] and [11], where the former [14] only has unlabeled target data and the latter [11] only has limited labeled target data. This paper seems to be a mixed setting of the above two settings. In this sense, the reviewer has the following three concerns:

(1).  with both labeled and unlabeled target data, existing semi-supervised learning methods (here I give a reference example [ref1]. The authors may want to check the more recent SOTA methods). The authors should consider to show the superiority of the current paper over this kind of methods.

[ref1] Kostopoulos, Georgios, et al. "Semi-supervised regression: A recent review." Journal of Intelligent & Fuzzy Systems 35.2 (2018): 1483-1500.

(2). as the paper mixed the setting of [11] and [14], the reviewer expects the method should clearly outperforms [11] and [14] as it uses more target information. However, from Tabel 3, the methods is not as good as [11] at the task M -> Mu.

(3). How the scale of labelled target data affects the final performance? Furthermore, how the ratio of labelled and unlabeled target data matter?

4.The empirical evaluations are not very convincing. In Table 1 and 2, the DINO is only slightly better than those baselines. In table 3, DINO is not as good as [11]. Note that [11] is a 12-years ago paper, the authors should consider more recent adaptive GP methods for comparisons.



######################### After rebuttal ##################
I would like to thank the authors' detailed response, which well address my concerns. I am willing to increase my score accordingly.



**Ethics Review Area:**

["I don’t know"]

**Limitations:**

The limiatation is stated in Appendix, mainly for the possible future working direction. It is more interesting to discuss the limitations on the technical part of the method.

**Strengths And Weaknesses:**

Overall, the paper is well-written. The idea of using domain adaptative Gaussian process to explain the neural network feature extractor is interesting. The distribution shift measure using the training dynamics of the distribution-informed neural network is a novel and sound point. However, I also have the following comments as stated below.

---

> ### Author Response · Authors · 2022-08-02
> **Response to Reviewer WMY4 (Part I)**
>
> Q1: The paper is specifically written for domain adaptation regression. However, to this reviewer, the technical contribution is not specific to regression. The authors claim that classification is naturally formulated as regression. Thus, the reviewer wonders how this method performs in those classification tasks (there are quite a lot SOTA UDA papers for classification, the authors may want to compare with some recent ones.).
>
> A1: We would like to point out that this is a misunderstanding. One of the crucial differences between domain adaptation regression and domain adaptation classification problems is the loss function. That is, one common loss function of domain adaptation regression is mean square error (MSE), whereas domain adaptation classification often uses cross-entropy loss with softmax. It has been revealed [14] that softmax changes the feature scales (i.e., Frobenius norm of feature matrix $||H||_F$ where $H$ is the hidden feature learned by the neural network), and the change of feature scales might lead to the performance degradation of domain adaptation (This is clarified in the related work, e.g., Lines 115-121 in the revised paper). Therefore, in this paper, we focus on the neural network with MSE loss for domain adaptation regression and use the more recent domain adaptation regression baselines in the experiments, e.g., RSD [14], TL-NTK [46], etc. The proposed method can be extended to domain adaptation classification when using cross-entropy loss with softmax. But the theoretical analysis, e.g., generalization and convergence, of our framework with cross-entropy bears significant difference, and it is out of the scope of this paper.
>
> Q2: To this reviewer, the related work section should be re-organized and further improved. Compared with discussing the big picture of domain adaptation, the authors should directly focus on the regression works on domain adaptation or transfer learning. The authors may want to add two more paragraphs, one discussing domain adaptation regression works and another one discussing the GP based adaptation setting, as these two research lines are closely related to the current paper. More importantly, it is more convincing to discuss how those closely related works, e.g., [14] and [11], differ from the current paper, instead of just listing them out.
>
> A2: One major contribution of this paper is to show the connections between the proposed DINO framework and previous popular domain adaptation frameworks, including domain invariant representation learning, reweighting and adaptive Gaussian process. That is why we would like to provide the background and related works for previous frameworks in Subsection 2.1.
>
> In addition, all those three frameworks can be applied for domain adaptation regression, e.g., [14] for domain invariant representation learning, [23] for reweighting, and [11] for adaptive Gaussian process. [14] proposed a representation subspace distance to measure the distribution shift for domain adaptation regression. But no theoretical analysis regarding the convergence and generalization bound for this approach is provided. In contrast, in this paper, we use the dynamics of a novel distribution-informed neural network to measure the distribution shift. We illustrate in Lines 252-259 that this discrepancy measure can not only unify the domain adaptation regression framework with neural networks, but also enable the theoretical convergence and generalization analysis. As shown in Subsection 3.2.3, [11] used an adaptive kernel with a hyper-parameter τ indicating the relatedness of source and target domains. But this heuristic adaptive kernel is not explained in [11]. Instead, in our paper, we show that the relatedness of domains can be explained as a distribution kernel identifying the similarity of the input-oriented distributions of source and target domains (see Lemma 4.1). In the revised version, we add more discussion on the related domain adaptation regression works (see Lines 115-126).

---

> > ### Author Response · Authors · 2022-08-02
> > **Response to Reviewer WMY4 (Part II)**
> >
> > Q3: To this reviewer, the setting studied in this paper is different from the settings of [14] and [11], where the former [14] only has unlabeled target data and the latter [11] only has limited labeled target data. This paper seems to be a mixed setting of the above two settings. In this sense, the reviewer has the following three concerns:
> >
> > Q3-1: with both labeled and unlabeled target data, existing semi-supervised learning methods (here I give a reference example [ref1]. The authors may want to check the more recent SOTA methods). The authors should consider to show the superiority of the current paper over this kind of methods.
> > [ref1] Kostopoulos, Georgios, et al. "Semi-supervised regression: A recent review." Journal of Intelligent & Fuzzy Systems 35.2 (2018): 1483-1500.
> >
> > A3-1: We would like to point out that domain adaptation typically assumes there is little or no label information from the target domain. Specifically, the problem setting with limited labeled target examples and adequate unlabeled target data has been widely studied in previous works [8,9,18]. The proposed DINO-INIT can be applied without using any unlabeled target data (the same as [11]), while both DINO-INIT and DINO-Train can exploit limited labeled and adequate unlabeled target data. The major difference between our problem setting and semi-supervised learning is that semi-supervised learning methods do not consider the knowledge transfer from the source domain, thus leading to sub-optimal performance in our setting. Moreover, it is also flexible to add existing semi-supervised regularizations over the target data into our domain adaptation framework. Since our paper focuses more on knowledge transferability across domains, we would like to leave the incorporation of our framework and semi-supervised learning techniques as our future work.
> >
> > Q3-2: as the paper mixed the setting of [11] and [14], the reviewer expects the method should clearly outperforms [11] and [14] as it uses more target information. However, from Tabel 3, the methods is not as good as [11] at the task M -> Mu.
> >
> > A3-2: We would like to point out that in most cases of Tables 1-3, our DINO algorithms significantly outperform [11] as well as more recent adaptive GP method [46].
> >
> > Q3-3: How the scale of labelled target data affects the final performance? Furthermore, how the ratio of labelled and unlabeled target data matter?
> >
> > A3-3: Theorem 4.5 shows that the expected target error is bounded by $O(1/\sqrt{n^l_{tgt}})$, where $n^l_{tgt}$ is the number of labeled target examples. That is, the model performance can be improved by using more labeled target examples. In addition, we use all the target examples to estimate the domain discrepancy, so the ratio of labeled and unlabeled target data would not affect the estimation of domain discrepancy (see the third term in Theorem 4.5). The ratio of labeled and unlabeled target data would only affect the second term of Theorem 4.5 as well as the sample complexity $O(1/\sqrt{n^l_{tgt}})$. This tells us that more labeled target data can help make the generalization error bound tighter and thus improve the model performance.
> >
> > Q4: The empirical evaluations are not very convincing. In Table 1 and 2, the DINO is only slightly better than those baselines. In table 3, DINO is not as good as [11]. Note that [11] is a 12-years ago paper, the authors should consider more recent adaptive GP methods for comparisons.
> >
> > A4: We would like to point out that this is a misunderstanding. We did compare our algorithms with the recent adaptive GP method TL-NTK [Maddox et al., 2021] in our experiments. Our results in Tables 1-3 show that our algorithms can outperform TL-NTK by a large margin in most cases. For example, compared to TL-NTK, the mean absolute error of DINO decreases by over 20% on dSprites ($C \rightarrow N$) in Table 1.

---

### Official Review · Reviewer_24HZ · 2022-07-11

**Rating:** 5
**Confidence:** 4
**Soundness:** 3 good
**Presentation:** 1 poor
**Contribution:** 3 good

**Summary:**

This paper proposes a domain adaptation method through a learning algorithm that knows the distribution of source and target well. The proposed method performs regression task when the distribution of source and target are different. Their learning algorithm consider maximizing RKHS mean discrepancy. Various experiments show that the proposed method improve the performance of the previous methods.

**Questions:**

Q1) A lot of domain adaptation algorithms and models have been studied in the classification task. Why are these algorithms and models not applicable to regression task?
Why is the proposed method particularly suitable for regression problems?

Q2) In fact, domain adaptation for classification can be easily applied to regression problems. Can you do a performance comparison with them in regression task?

Q3) It is recommended to draw a workflow figure that shows everything from input to output at once. Can you draw a figure to understand the concept of the proposed method?

**Limitations:**

The authors explain the societal impact and limitation of the paper well.

**Strengths And Weaknesses:**

Strengths
+) Motivation and the description of the proposed algorithm is convincing.
+) Various experiments are performed, and the performance of the proposed method is better than previous regression DA methods.

Weaknesses
-) The paper representation needs to be improved. There are too many formulas in the main paper. Except for essential formulas, authors need to pass them all to the appendix and explain the concept and analysis in the main paper.
-) There are not enough SOTA methods in the experimental results. In particular, AT-GP was proposed in 2010 and the performance improvement of the proposed method is not remarkable.

---

> ### Author Response · Authors · 2022-08-02
> **Response to Reviewer 24HZ**
>
> Q1: The paper representation needs to be improved. There are too many formulas in the main paper. Except for essential formulas, authors need to pass them all to the appendix and explain the concept and analysis in the main paper. -) There are not enough SOTA methods in the experimental results. In particular, AT-GP was proposed in 2010 and the performance improvement of the proposed method is not remarkable.
>
> A1: We would like to clarify that the formulas in the main paper are mainly used to explain the crucial idea and concept of our algorithms and analysis. It might be much clearer to show how our algorithms (as well as the motivations behind them) work using formulas, compared to explaining them in plain words. In addition to AT-GP, we did use other recent baselines, including NNGP [Lee et al., 2018] and NTKGP [He et al., 2020], TL-NTK [Maddox et al., 2021], DAN [Long et al., 2015], WANN [de Mathelin et al., 2020] and RSD [Chen et al., 2021]. It is notable that both AT-GP and TL-NTK [Maddox et al., 2021] are Gaussian process based domain adaptation approaches. Compared to other two frameworks (i.e., domain invariant representation and reweighting), Gaussian process-based approaches attract less attention in recent years. AT-GP is a representative baseline for this type of baselines.
>
> Q2: A lot of domain adaptation algorithms and models have been studied in the classification task. Why are these algorithms and models not applicable to regression task? Why is the proposed method particularly suitable for regression problems?
>
> A2: We would like to clarify the difference between domain adaptation regression and domain adaptation classification problems as follows. One crucial difference of these problems is the loss function. That is, one common loss function of domain adaptation regression is mean square error (MSE), whereas domain adaptation classification often uses cross-entropy loss with softmax. It has been revealed [14] that softmax changes the feature scales (i.e., Frobenius norm of feature matrix $||H||_F$ where $H$ is the hidden feature learned by the neural network), and the change of feature scales might lead to the performance degradation of domain adaptation (This is clarified in the related work, e.g., Lines 115-121 in the revised paper). Therefore, in this paper, we focus on the neural network with MSE loss for domain adaptation regression. The proposed method can be extended to domain adaptation classification when using cross-entropy loss with softmax. But the theoretical analysis, e.g., generalization and convergence, of our framework with cross-entropy bears significant difference, and it is out of the scope of this paper.
>
> Q3: In fact, domain adaptation for classification can be easily applied to regression problems. Can you do a performance comparison with them in regression task?
>
> A3: We would like to clarify that in our experiments, we did use the domain adaptation classification baselines DAN [45], WANN [23]. As introduced in Subsection 5.1, WANN [23] reweights the source examples and DAN [45] learns the domain invariant representation. To make them work for domain adaptation regression tasks, following [14], we replace the cross-entropy loss of these baselines with mean square error.
>
> Q4: It is recommended to draw a workflow figure that shows everything from input to output at once. Can you draw a figure to understand the concept of the proposed method?
>
> A4: As mentioned in Line 228 and Line 267, the proposed algorithms are summarized in Algorithm 1 and Algorithm 2 (see Appendix A.10). It clearly shows the inputs, outputs, and training procedures of our algorithms.

---

### Official Review · Reviewer_cqod · 2022-07-11

**Rating:** 5
**Confidence:** 2
**Soundness:** 3 good
**Presentation:** 2 fair
**Contribution:** 3 good

**Summary:**

This paper proposes a domain adaptation (DA) regression framework based on the distribution-informed neural network, which connects to previous techniques, including domain invariant learning, reweighting, and adaptive Gaussian process. It also theoretically analyzes the convergence and generalization bound of the framework. The method is evaluated on three DA regression datasets and outperforms some state-of-the-art methods.

**Questions:**

Besides the points listed above, I have some questions:
1. Can you clarify why the method proposed in section 4.2 is called DINO-Init? How does it relate to section 4.3? In experiments, I find that DINO-Init and DINO-Train may outperform each other in different cases. Can you explain this? How could we choose between these two algorithms in practice?
2. Can you clarify what the ‘training dynamics’ mean in Line 251? Why is it called ‘training dynamics’? Why is it better than previous measurements?
3. In Line 238, why do we use all examples as the basis examples? Would this prevent the method from scaling to larger datasets? How do the different choices of basis example sets influence the method?
4. In practice, how should we choose hyper-parameters such as \alpha and \mu in Equation 7?

**Limitations:**

The authors addressed the limitation that the framework may not handle multi-source settings. I think another limitation is that it is unknown whether this paper can generalize to commonly-used deeper networks with diverse architectures.

**Strengths And Weaknesses:**

Strengths:
1. It is interesting and inspiring to explore the ideas of distribution-informed neural networks in the DA regression problem.
2. This paper provides theoretical analyses on the convergence and generalization of the proposed DA regression framework. Different from previous theories, distribution shifts are measured using the training dynamics of the neural network.
3. The proposed method outperforms some state-of-the-art DA regression methods on three benchmarks.

Weaknesses:
1. I think this paper is generally not clearly written and hard to follow, especially the most important sections 4.2 and 4.3. The titles of the subsections, such as ‘4.2 Initialization’, are confusing. I think it also needs more transitional sentences to lead or summarize the sections and to clarify the motivations.
2. The proposed framework is mainly evaluated on a shallow fully-connected neural network. It is unknown whether it can be extended to other architectures (such as CNN or Transformers), deeper models, and models already pre-trained from large datasets since these models may be more commonly used in practice.
3. The method is mainly evaluated on some toy datasets, and I think it needs some experiments on more realistic situations.

---

> ### Author Response · Authors · 2022-08-02
> **Response to Reviewer cqod (Part I)**
>
> Q1: I think this paper is generally not clearly written and hard to follow, especially the most important sections 4.2 and 4.3. The titles of the subsections, such as ‘4.2 Initialization’, are confusing. I think it also needs more transitional sentences to lead or summarize the sections and to clarify the motivations.
>
> A1: We would like to clarify that the titles “Initialization” and “Gradient Descent Training” indicate two different statuses of the proposed distribution-informed neural network (introduced in Subsection 4.1). To be more specific, “Initialization” indicates that the model parameters of the proposed distribution-informed neural network are randomly initialized. “Gradient Descent Training” means that we use gradient descent to update the model parameters of the proposed distribution-informed neural network. Those two statuses motivate us to develop our algorithms DINO-INIT and DINO-TRAIN. In the revised paper, we have added transitional sentences to clarify those approaches in Lines 212-214.
>
> Q2: The proposed framework is mainly evaluated on a shallow fully-connected neural network. It is unknown whether it can be extended to other architectures (such as CNN or Transformers), deeper models, and models already pre-trained from large datasets since these models may be more commonly used in practice.
>
> A2: We would like to point out that the proposed framework can be naturally used on deeper models, because both DINO algorithms and their theoretical analysis have no constraint on the network depth. Besides, the proposed DINO framework can be easily generalized to other network architectures. This is because previous works [5, 29, 65, 66] have shown the existence of NNGP and NTK in different network architectures, including (residual) convolutional neural networks, recurrent neural networks, transformer, etc. This enables us to adapt the proposed DINO framework to different network architectures by developing the distribution-informed NNGP and NTK correspondingly. We have clarified it in Appendix A.10 in the revised version. Finally, pre-trained models cannot be used in our framework. That is because our theoretical analysis and proposed framework consider the networks in the limit of infinite network width. Most existing pre-trained models do not satisfy this condition.
>
> Q3: The method is mainly evaluated on some toy datasets, and I think it needs some experiments on more realistic situations.
>
> A3: We are afraid that this is a misunderstanding. Following previous work [14], we did use the domain adaptation regression benchmarks dSprites and MPI3D to evaluate the proposed algorithms. Besides, we also used another realistic agriculture data sets with two domains: Maize (M) and Maize_UNL (MU). It collects the biochemical properties (e.g., leaf hyperspectral reflectance) of maize from different cities. The task is to predict the Nitrogen content of maize. This realistic data set also helps us evaluate the proposed algorithms.
>
> Q4: Can you clarify why the method proposed in section 4.2 is called DINO-Init? How does it relate to section 4.3? In experiments, I find that DINO-Init and DINO-Train may outperform each other in different cases. Can you explain this? How could we choose between these two algorithms in practice?
>
> A4: (1) In Subsection 4.1, we present the proposed distribution-informed neural network. Then in the next Subsections 4.2 and 4.3, we analyze how this distribution-informed neural network can perform domain adaptation at different statuses. Specifically, in Subsections 4.2, we assume that the model parameters of the proposed distribution-informed neural network are randomly initialized. We show in Lemma 4.1 that the output of the distribution-informed neural network at initialization is a Gaussian process. That is why we call the derived method “DINO-Init” in this case. In the revised paper, we provide a detailed description on Lines 212-214.
>
> (2) As explained in Appendix A.10, it is empirically observed [41] that for standard neural networks (e.g., fully-connected neural networks), NNGP might outperform NTK in some cases. Their performance varies with different model architectures and data sets. One explanation in previous work [32] is that the covariance of Gaussian process induced by NTK is much larger than that induced by NNGP, i.e., $\Sigma_{NTKGP} \succeq \Sigma_{NNGP}$ (or $\Sigma_{NTKGP} - \Sigma_{NNGP}$ is a positive definite kernel). This might lead to an unstable prediction function of the Gaussian process induced by NTK. We did have similar observations for the proposed distribution-informed neural network, e.g., DINO-INIT over adaptive NNGP and DINO-TRAIN over distribution-informed NTK. Both algorithms can be applied in practice.

---

> > ### Author Response · Authors · 2022-08-02
> > **Response to Reviewer cqod (Part II)**
> >
> > Q5: Can you clarify what the ‘training dynamics’ mean in Line 251? Why is it called ‘training dynamics’? Why is it better than previous measurements?
> >
> > A5: As explained in Lines 67-69, “training dynamics” indicate how the distribution-informed neural network is changed during model training with gradient descent. In [38], the dynamics of a neural network are used to analyze the model convergence. In our paper, we explore the dynamics of the distribution-informed neural network to derive the theoretical convergence and generalization bound of our framework. Besides, in Eq. (8), we also use “training dynamics” of distribution-informed neural network over different domains to measure the distribution shift. The benefits of this measure have been illustrated in Lines 268-276. That is, compared to previous works, our discrepancy measure Eq. (8) can not only unify the domain adaptation regression framework with neural networks, but also enable the theoretical convergence and generalization analysis. In contrast, previous measurements tend to use a two-stage method, and their performance is largely affected by the pre-defined latent feature space. This is also confirmed in our experiments (see Subsection 5.2 and Figure 3(b)).
> >
> > Q6: In Line 238, why do we use all examples as the basis examples? Would this prevent the method from scaling to larger datasets? How do the different choices of basis example sets influence the method?
> >
> > A6: The reason why we choose all the basis examples is that more basis examples can help represent the NNGP kernel space. We show in Eq. (4) that the input-oriented distribution feature representation relies on the subspace spanned by the feature maps of basis examples. As a result, more basis examples might enlarge this subspace. It is notable that this subspace is determined by linearly independent feature maps. Thus, we can also identify the linearly independent feature maps in the NNGP kernel space, in order to reduce the number of basis examples in real scenarios. But it would be time-consuming due to the high dimensionality of NNGP kernel space.
> >
> > Q7: In practice, how should we choose hyper-parameters such as \alpha and \mu in Equation 7?
> >
> > A7: In the experiments, we have labeled target examples for model training. Thus, we choose the hyper-parameters based on the model performance on those labeled target examples.

---

> > > ### Comment · Reviewer_cqod · 2022-08-09
> > > **Response to Rebuttal**
> > >
> > > Thanks for your detailed responses, which have addressed most of my concerns.
> > >
> > > An additional question on Q6: Would choosing all the basis examples prevent the method from scaling to larger datasets?
> > >
> > > Besides, it seems to be a small limitation that:
> > >
> > > - The performance on other architectures is not verified by experiments.
> > >
> > > - Pre-trained models cannot be used in the framework, while pre-training and fine-tuning is a popular and common practice in many machine learning applications.
> > >
> > > Overall, I think it is an interesting paper and I would like to keep my current score.

---

> > > > ### Author Response · Authors · 2022-08-09
> > > > **Response to Reviewer cqod**
> > > >
> > > > Thank you very much for your comments.
> > > >
> > > > Q1: Would choosing all the basis examples prevent the method from scaling to larger datasets?
> > > > A1: There are two intuitive methods to help the proposed method scale to large data sets. One is that as shown in our previous answer, it can detect linearly independent feature maps from all the basis examples. It can also find the optimal basis examples in this case, but it is time-consuming. The other one is to randomly choose the subset of basis examples. It is computationally efficient, but it might lead to suboptimal selection.
> > > >
> > > > Q2: The performance on other architectures is not verified by experiments.
> > > > A2: In this paper, we focus more on developing the domain adaptation regression framework based on the distribution-informed neural networks. The architecture comparison of NTK and NNGP has been studied in previous work [41]. Thus, we did not pay much attention to empirically comparing different architectures in our experiments.
> > > >
> > > > Q3: Pre-trained models cannot be used in the framework, while pre-training and fine-tuning is a popular and common practice in many machine learning applications.
> > > > A3: We would like to point out that pre-training/fine-tuning is a different learning paradigm from the proposed framework in this paper. The major differences are as follows. (1) Pre-training/fine-tuning actually learns different models (i.e., obtain a source pre-trained model, then fine-tune on target data to obtain target model) for source and target domains. In contrast, we assume the same hypothesis for both domains. (2) As explained in our previous answers, our theoretical analysis and proposed framework consider the networks in the limit of infinite network width. Most existing pre-trained models do not satisfy this condition. This is why our derived theoretical results and algorithms can not be directly applied to the pre-training and fine-tuning paradigm. The theoretical generalization and convergence of this pre-training/fine-tuning paradigm are also underexplored. But it is out of the scope of this paper.

---

### Official Review · Reviewer_ezF7 · 2022-07-11

**Rating:** 5
**Confidence:** 1
**Soundness:** 3 good
**Presentation:** 3 good
**Contribution:** 3 good

**Summary:**

This paper proposes a framework called DINO for domain adaptation regression, which is based on distribution-informed neural networks. This framework can be theoretically related to a) domain invariant learning, b) source data reweighting, and c) adpative gaussian processes.  Authors show that convergence of such networks can be guaranteed. Also, the framework uses the training dynamics of the network when measuring the distribution shift between domains. Experiments were conducted on dSprites, MPI3d, and Plant Phenotyping - two of which are image datasets. Overall, both DINO-INIT and DINO-TRAIN show strong performances in all three datasets.

**Questions:**

* In this paper, the authors use an L-layer fully-connected network for all experiments. Does the theoretical analysis and method presented in this paper naturally extend to diverse architectures, such as CNNs and/or transformers?
* It seems like one underlying assumption is that the neural network should be over-parametrized. But can we say that a 6-layer fully-connected network is over-parametrized for dSprites or MPI3D, both of which are image datasets?

**Limitations:**

Limitations are mentioned in the Appendix. Authors note that the framework is based on single-source domain adaptation.

**Strengths And Weaknesses:**

**Strengths**
* DINO presents a unified framework that is well connected with a few previous works. In other words, the authors show that methods proposed in other works may be interpreted as a special instantiation of the DINO framework proposed in this paper.
* The authors theoretically verify that the proposed framework is able to converge.
* Overall, the paper is well written.

**Weaknesses**
* not clear what the difference between DINO-INIT and DINO-TRAIN is. First mention of DINO is in the intro, while the second mention is in Section 5. Explanation of DINO-INIT and DINO-TRAIN are in the Appendix (A.10), but should really be part of the main paper (or at least, it should be mentioned that DINO-INIT/TRAIN are explained in the Appendix).
* Why is it that DINO-INIT outperforms DINO-TRAIN in both dSprites and MPI3D? More insights could be provided for this.
* I think the paper could be improved by reducing the amount of theoretical analysis in the main paper (could be moved to the Appendix), as I feel that it takes away from the main points of the paper.
* It is not clear how DINO would be generalized to more practical datasets/network architectures. Overall, experiments that demonstrate practicality would make the submission much stronger.

**Other**
* L258 Typo: (8) can **not** only unify ...
* Some equations leak into the margins. For example, equation after Lines 229, 285, and 291.

---

> ### Author Response · Authors · 2022-08-02
> **Response to Reviewer ezF7**
>
> Q1: not clear what the difference between DINO-INIT and DINO-TRAIN is. First mention of DINO is in the intro, while the second mention is in Section 5. Explanation of DINO-INIT and DINO-TRAIN are in the Appendix (A.10), but should really be part of the main paper (or at least, it should be mentioned that DINO-INIT/TRAIN are explained in the Appendix).
>
> A1: Thank you for your suggestions. We would like to clarify that DINO-INIT is developed based on the distribution-informed neural network at initialization (Subsection 4.2 and Algorithm 1 in Appendix A.10), and DINO-TRAIN algorithm is given by the distribution-informed neural network with gradient descent training (Subsection 4.3 and Algorithm 2 in Appendix A.10). We have clarified their difference in the revised version in Lines 212-217 and Lines 258-259.
>
> Q2: Why is it that DINO-INIT outperforms DINO-TRAIN in both dSprites and MPI3D? More insights could be provided for this.
>
> A2: As explained in Appendix A.10, it is empirically observed [41] that for standard neural networks (e.g., fully-connected neural networks), NNGP might outperform NTK in some cases. Their performance varies with different model architectures and data sets. One explanation in previous work [32] is that the covariance matrix of Gaussian process induced by NTK is much larger than that induced by NNGP, i.e., $\Sigma_{NTKGP} \succeq \Sigma_{NNGP}$ (or $\Sigma_{NTKGP} - \Sigma_{NNGP}$ is a positive definite kernel). This might lead to an unstable prediction function of the Gaussian process induced by NTK. We have similar observations for the proposed distribution-informed neural network, e.g., DINO-INIT over adaptive NNGP and DINO-TRAIN over distribution-informed NTK. We have added the analysis in Lines 368-370 in the revised paper.
>
> Q3: I think the paper could be improved by reducing the amount of theoretical analysis in the main paper (could be moved to the Appendix), as I feel that it takes away from the main points of the paper.
>
> A3: We would like to clarify that we provide the theoretical analysis from two aspects. One illustrated in Subsections 4.2&4.3 is the convergence and generalization (see Lemma 4.1, Theorems 4.4 and 4.5) of the proposed algorithms. The other one illustrated in Subsection 4.4 is how the proposed algorithms are connected to the existing adaptive Gaussian process (see Corollary 4.6), reweighting (see Corollary 4.7) and invariant representation learning (see Corollary 4.8). In the revised version, we move some theoretical results to the appendix (see Theorem A.1).
>
> Q4: It is not clear how DINO would be generalized to more practical datasets/network architectures. Overall, experiments that demonstrate practicality would make the submission much stronger.
>
> A4: We would like to point out that the proposed DINO framework can be easily generalized to other network architectures. This is because previous works [5, 29, 65] have shown the existence of NNGP and NTK in different network architectures, including (residual) convolutional neural networks, recurrent neural networks, etc. Therefore, we can adapt our algorithms and theoretical analysis to those network architectures as well. In this paper, we focus on the most used fully connected network, and the exploration of network architecture comparison in our framework is beyond the scope of this paper. We have clarified it in Appendix A.10 in the revised version.
>
> In addition, we also fix the typo and equation leakage issues in the revised paper.

---

> > ### Comment · Reviewer_ezF7 · 2022-08-09
> > **Thank you for the response**
> >
> > Dear Authors,
> >
> > First, I would like to sincerely apologize for taking so long to get back to you. I do not expect the authors to respond to such a late reply to the rebuttal.
> >
> > I have read the rebuttal and checked the revised paper. Thank you for the changes and clarifications.
> >
> > I still have concerns regarding the scalability of such method to modern architectures. In A4, the authors do state that the existence of NNGP and NTK in CNNs and RNNs has been shown in previous works and that exploration of network architectures is beyond the scope of this paper. However, I still think that one simple experiment, even with a small CNN (on MPI3D or dSprites, since both are image datasets), could be extremely valuable in terms of validating the statement above. For example, by doing so, authors could gain/provide insights into the some obstacles faced while extending their method beyond MLPs. And, perhaps, this extension is not as simple as it seems.
> >
> > Anyways, I would like to apologize again for the late reply - I understand how frustrating it must have been.
> >
> > As for the score, I am leaning towards raising my score, but as you already know, my confidence score is very low. Since we have Reviewer-AC discussions coming up tomorrow, please give me a bit more time to go over the paper again and finalize my score.
> >
> > Best,
> > Reviewer ezF7

---

> > > ### Author Response · Authors · 2022-08-09
> > > **Response to Reviewer ezF7**
> > >
> > > Thank you very much for your suggestions. Due to the limited time, we might not have the experimental results for validating the proposed algorithms based on CNNs. But we found that previous work [41] has a similar empirical evaluation of the neural networks based on Fully-Connected (FCN) and Convolutional (CNN) networks for standard supervised learning. Similarly, we will add the empirical analysis of our proposed algorithms based on CNNs in the revised version.

---

### Meta-Review · Area_Chair_h8Zo · 2022-08-27

**Recommendation:** Accept
**Confidence:** Less certain

**Metareview:**

This paper works towards domain adaptation in the regression setting through the distribution-informed neural networks. It provides a nice theoretical elaboration to the three mainstream methods in this field: domain invariant learning, source data re-weighting, and adaptive Gaussian processes. However, reviewers were on the borderline even after the author rebuttal and after the rating increase. AC carefully justified the merits and flaws of this paper and felt that the former slightly outweighs the latter, and this is in agreement with the SAC. The major concern is the use of fully-connected network for theoretical analysis and empirical evaluation --- the scalability of the theory to modern neural networks seems not easy while the paper does not provide sufficient evidence in this regard. However, considering this paper is more theory-oriented, such a concern is down-weighted a little bit. Thus the paper is recommended for acceptance. Authors are suggested to improve their paper by incorporating the rebuttal and further addressing the reviews, in particular improving the clarity and smoothness of the presentation to be more readable.

**Award:**

No

---

### Decision · Program_Chairs · 2022-09-14

Accept